# Theoretically Grounded Framework for LLM Watermarking: A Distribution-Adaptive Approach

**Haiyun He**[*]
HKUST (GZ)
Guangzhou, China
haiyunhe@hkust-gz.edu.cn

**Yepeng Liu**[*]
UC Santa Barbara
Santa Barbara, CA, USA
yepengliu@ucsb.edu

**Ziqiao Wang**
Tongji University
Shanghai, China
ziqiaowang@tongji.edu.cn

**Yongyi Mao**
University of Ottawa
Ottawa, ON, Canada
ymao@uottawa.ca

**Yuheng Bu**
UC Santa Barbara
Santa Barbara, CA, USA
buyuheng@ucsb.edu

## Abstract

Watermarking has emerged as a crucial method to distinguish AI-generated text from human-created text. Current watermarking approaches often lack formal optimality guarantees or address the scheme and detector design separately. In this paper, we introduce a novel, unified theoretical framework for watermarking Large Language Models (LLMs) that jointly optimizes both the watermarking scheme and detector. Our approach aims to maximize detection performance while maintaining control over the worst-case false positive rate (FPR) and distortion on text quality. We derive closed-form optimal solutions for this joint design and characterize the fundamental trade-off between watermark detectability and distortion. Notably, we reveal that the optimal watermarking schemes should be adaptive to the LLM's generative distribution. Building on our theoretical insights, we propose a distortion-free, distribution-adaptive watermarking algorithm (DAWA) that leverages a surrogate model for model-agnosticism and efficiency. Experiments on Llama2-13B and Mistral-$8\times$7B models confirm the effectiveness of our approach, particularly at ultra-low FPRs. Our code is available at `https://github.com/yepengliu/DAWA`.

## 1 Introduction

Arising with Large Language Models (LLMs) [1] is a double-edged sword: while they boost productivity, they also introduce new risks, including plagiarism, challenges to content accountability, and other forms of misuse. Watermarking, a hidden and machine-verifiable tag inserted into LLMs' outputs, has therefore become a critical line of defense for publishers, educators, and regulators.

Existing watermarking techniques for AI-generated text are commonly grouped into two main categories: post-process and in-process [2, 3]. Post-process watermarking is applied after the text is generated [4–16], while in-process watermarking embeds watermarks during generation [17–36]. There is a growing interest in in-process methods due to their invisibility, flexibility, and seamless integration with negligible latency. For additional related works, please refer to Appendix A.

However, realizing the full potential of in-process watermarking requires careful design. A key challenge is controlling the false positive rate (FPR), as even a single error can lead to serious

---

[*]These authors contributed equally to this work.

39th Conference on Neural Information Processing Systems (NeurIPS 2025).

consequences, such as wrongly accusing a human author. Maintaining a high true positive rate (TPR) while keeping an ultra-low FPR, e.g., $1\mathrm{e}{-}05$, is therefore essential. In addition, effective in-process watermarking should be both **detectable** and **distortion-free**: the watermark must be reliably identified under strict false positive rate (FPR) constraints, while preserving the quality and distribution of the original output [35, 37]. Moreover, a practical detector should be **model-agnostic**, operable without access to the original LLM or its prompt [25]. Balancing all these goals is challenging, and existing methods often fall short in one or more dimensions.

Many heuristic designs typically embed watermarks into generated tokens by perturbing the token logits (e.g., the green-red list [25]) or modifying the sampling process (e.g., Gumbel-Max sampling [38]). Detection is usually performed using handcrafted score statistics. However, these "trial-and-error" approaches rely heavily on empirical tuning, with no formal optimality guarantees.

In principle, designing a watermarking system can be formulated as a constrained optimization problem: maximizing the detection probability TPR with the FPR and text distortion under control. Recent theoretical efforts have taken steps towards this goal. For instance, Takezawa et al. [39], Wouters [40], and Cai et al. [41] focus on optimizing the logit perturbation strategy for the green-red list watermarking scheme. Huang et al. [42] frame watermarking as a statistical independence test between text and watermark and derive the optimal scheme for a *fixed detector*, but stop short of a practical, model-agnostic algorithm. On the other hand, Li et al. [43] optimizes the detector for a *fixed watermarking scheme* using i.i.d. pivotal statistics. Consequently, these approaches do not guarantee overall system optimality. A significant gap in these theoretical explorations is that none of them consider the *joint optimization* of both the watermarking scheme and the detector.

Specifically, both the way the watermark is embedded and the type of signal used can be optimized. However, existing approaches often rely on fixed, simplistic designs, such as using randomly generated bits or samples from uniform distributions, leaving much of the design space unexplored. These restrictive choices may prevent existing schemes from achieving optimal performance.

This paper aims to fill the gap. We develop a **novel, unified theoretical framework** that subsumes most existing in-process schemes, aiming to *jointly optimize* the watermarking-detector pair for any token-sequence length $T$ that achieves the *best trade-off* between watermark detectability and text distortion. Unlike the classical watermarking paradigm, which employs fixed watermark distributions, our framework generalizes watermarking to an *adaptive setting* where the watermark signal exploits the LLM's generative distribution. This opens up one more degree of freedom to optimize the sampling distributions of watermark signals, thereby enhancing detection reliability at ultra-low FPRs, as shown in Figure 1.

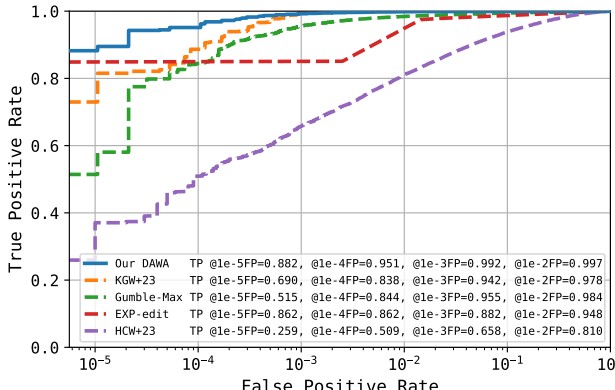

Figure 1: Comparison of TPR at ultra-low FPR among different watermarking methods.

Our contributions can be summarized as follows:

- In Section 2, we propose a unified theoretical framework for LLM watermarking and detection that encompasses most existing watermarking methods. This framework features a common randomness shared between watermark generation and detection to perform an independence test.

- In Section 3, we characterize the universally minimum Type-II error (i.e., $1-\text{TPR}$) as a function of FPR and text distortion level, revealing a fundamental trade-off between **detectability and distortion**. More importantly, we identify the closed-form jointly optimal solutions for watermarking schemes and detectors, providing a guideline for practical design, i.e., watermarking schemes should adapt to the generative distribution of LLMs.

- In Section 4, we translate the sequence-level optimum to a practical token-level watermarking design. We prove that it retains reliable detection performance and is inherently robust against token replacement attacks. In Section 5, we introduce **DAWA** (**D**istribution-**A**daptive **W**atermarking **A**lgorithm), a distortion-free implementation of the token-level design, which leverages a sur-

rogate language model and the Gumbel-Max sampling trick to achieve **model-agnosticism** and computational efficiency.

- In Section 6, we conduct extensive experiments on Llama2-13B [1] and Mistral-8×7B [44], across multiple datasets. DAWA consistently outperforms the compared methods, even under token replacement attacks, and maintains high text quality. As shown in Figure 1, DAWA achieves superior detection capabilities at ultra-low FPRs.

- Lastly, we sketch in Appendix J how to extend our theoretical framework to semantic-invariant watermarking removal attacks and derive the associated detectability–distortion–robustness trade-off, guiding future semantic-based watermark designs.

## 2 Preliminaries and Problem Formulation

**Notations.** For a sequence of random variables $X_1, \ldots, X_n$, and any $i, j \in [n]$ with $i \leq j$, we denote $X_i^j := (X_i, \ldots, X_j)$. We may use distortion function, namely, a function $\mathsf{D} : \mathcal{P}(\mathcal{X}) \times \mathcal{P}(\mathcal{X}) \to [0, +\infty)$ to measure the dissimilarity between two distributions in $\mathcal{P}(\mathcal{X})$. For example, the total variation distance, as a distortion, between $\mu, \nu \in \mathcal{P}(\mathcal{X})$ is $\mathsf{D}_{\mathsf{TV}}(\mu, \nu) := \int \frac{1}{2} |\frac{d\mu}{d\nu} - 1| \, d\nu$. For any set $A \subseteq \mathcal{X}$, we use $\delta_A$ to denote its indicator function, namely, $\delta_A(x) := \mathbb{1}\{x \in A\}$. Additionally, we denote $(x)_+ := \max\{x, 0\}$ and $x \wedge y := \min\{x, y\}$.

**Tokenization and NTP.** LLMs process text through "tokenization," namely, breaking it down into words or word fragments called "tokens." An LLM generates text token by token. Let $\mathcal{V}$ denote the token vocabulary, typically of size $|\mathcal{V}| = \mathcal{O}(10^4)$ [45–47, 1]. An *unwatermarked* LLM generates the next token $X_t$ based on a prompt pt and the previous tokens $x_1^{t-1}$ by sampling the Next-Token Prediction (NTP) distribution $Q_{X_t | x_1^{t-1}, \text{pt}}$. For simplicity, the prompt dependency is suppressed in notation throughout the paper. The joint distribution of a length-$T$ generated token sequence $X_1^T$ is then given by $Q_{X_1^T} := \prod_{t=1}^{T} Q_{X_t | X_1^{t-1}}$, which we assume to be identical to one that governs the human-generated text.

**A Framework for Watermarking Scheme.** Traditional post-hoc detectors identify AI-generated text by dividing the entire text space into rejection and acceptance regions, which relies on the assumption that certain sentences are unlikely to be written by humans. In contrast, modern LLM watermarking schemes achieve the same goal by analyzing the dependence structure between text $X_1^T$ and an auxiliary random sequence $\zeta_1^T$, thereby avoiding this restriction.

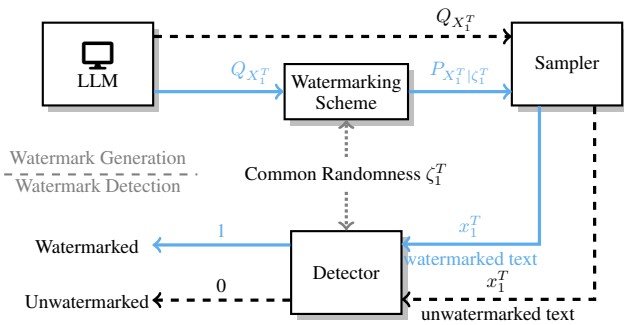

Figure 2: Overview of LLM watermarking and detection.

In this paper, we propose a general framework for LLM watermarking and detection, as shown in Figure 2, which encompasses most of the existing watermarking schemes. The watermarking scheme and detector share a common randomness represented by an *auxiliary random sequence* $\zeta_1^T$ drawn from a space $\mathcal{Z}^T$ (either discrete or continuous). After passing through a watermarking scheme, the watermarked LLM samples token sequence according to the modified NTP distribution $P_{X_t | x_1^{t-1}, \zeta_1^T}$, where $P_{X_1^T | \zeta_1^T} = \prod_{t=1}^{T} P_{X_t | X_1^{t-1}, \zeta_1^T}$. This process associates the generated text $X_1^T$ with an auxiliary sequence $\zeta_1^T$. Thus, the joint distribution of the watermarked token sequence $X_1^T$ is $P_{X_1^T}$, which might be different from the original $Q_{X_1^T}$. The detector can then distinguish whether the received sequence $X_1^T$ is watermarked or not based on the common randomness.

To evaluate the *distortion level* of a watermarking scheme, we measure the statistical divergence between the watermarked text distribution $P_{X_1^T}$ and the original one $Q_{X_1^T}$.

**Definition 1** ($\epsilon$-distorted watermarking scheme). *A watermarking scheme is $\epsilon$-distorted with respect to distortion* $\mathsf{D}$, *if* $\mathsf{D}(P_{X_1^T}, Q_{X_1^T}) \leq \epsilon$. *Here,* $\mathsf{D}$ *can be any distortion metric.*

Common examples of such divergences include squared distance, total variation, KL divergence, and Wasserstein distance. For $\epsilon = 0$, the watermarking scheme is *distortion-free*.

Specifically, our formulation allows the auxiliary random sequence $\zeta_1^T$ to take an arbitrary structure, which contrasts the rather restricted i.i.d. assumption considered in Li et al. [43, Working Hypothesis 2.1]. In practice, $\zeta_1^T$ is usually randomly generated using a shared `key` accessible during both watermark generation and detection. At first glance, our formulation may appear abstract, but its flexibility enables existing watermarking schemes to be interpreted as special cases within this framework.

**Example 1** (Existing watermarking schemes as special cases)**.** *In the Green-Red List watermarking scheme [25], at each position $t$, the vocabulary $\mathcal{V}$ is randomly split into a green list $\mathcal{G}$ and a red list $\mathcal{R}$, with $|\mathcal{G}| = \rho|\mathcal{V}|$ for some $\rho \in (0,1)$. This split is represented by a $|\mathcal{V}|$-dimensional binary auxiliary variable $\zeta_t$, indexed by $x \in \mathcal{V}$, where $\zeta_t(x) = 1$ means $x \in \mathcal{G}$; otherwise, $x \in \mathcal{R}$. The watermarking scheme is as follows:*

– *Compute a hash of the previous token $X_{t-1}$ using a hash function $h : \mathcal{V} \times \mathbb{R} \to \mathbb{R}$ and a shared secret `key`, i.e., $h(X_{t-1}, \text{key})$.*
– *Use $h(X_{t-1}, \text{key})$ as a seed to uniformly sample the auxiliary variable $\zeta_t$ from the set $\{\zeta \in \{0,1\}^{|\mathcal{V}|} : \|\zeta\|_1 = \rho|\mathcal{V}|\}$ to construct the green list $\mathcal{G}$.*
– *Sample $X_t$ from the adjusted NTP distribution which increases the logit of tokens in $\mathcal{G}$ by $\delta > 0$:*

$$P_{X_t|x_1^{t-1}, \zeta_t}(x) = \frac{Q_{X_t|x_1^{t-1}}(x) \exp(\delta \cdot \mathbb{1}\{\zeta_t(x)=1\})}{\sum_{x \in \mathcal{V}} Q_{X_t|x_1^{t-1}}(x) \exp(\delta \cdot \mathbb{1}\{\zeta_t(x)=1\})}.$$

*How our formulation encompasses several other watermarking schemes is provided in Appendix B.*

**Hypothesis Testing for Watermark Detection.** Note that a sequence $X_1^T$ generated by a watermarked LLM depends on $\zeta_1^T$, while $X_1^T$ and $\zeta_1^T$ are independent if written by humans. Therefore, detection involves distinguishing the following two hypotheses based on the pair $(X_1^T, \zeta_1^T)$:

• $H_0$: $X_1^T$ is generated by a human, i.e., $(X_1^T, \zeta_1^T) \sim Q_{X_1^T} \otimes P_{\zeta_1^T}$;

• $H_1$: $X_1^T$ is generated by a watermarked LLM, i.e., $(X_1^T, \zeta_1^T) \sim P_{X_1^T, \zeta_1^T}$.

We consider a model-agnostic detector $\gamma : \mathcal{V}^T \times \mathcal{Z}^T \to \{0,1\}$, which maps $(X_1^T, \zeta_1^T)$ to the hypothesis index (see Figure 2). In theory, we assume that the auxiliary sequence $\zeta_1^T$ can be fully recovered from $X_1^T$ and the common randomness, *while this assumption is dropped in practice.*

Detection performance is measured by the Type-I (false positive) and Type-II (false negative) errors:

$$\text{Type-I (i.e., FPR):} \quad \beta_0(\gamma, Q_{X_1^T}, P_{\zeta_1^T}) := \Pr(\gamma(X_1^T, \zeta_1^T) \neq 0 \mid H_0),$$
$$\text{Type-II (i.e., } 1-\text{TPR):} \quad \beta_1(\gamma, P_{X_1^T, \zeta_1^T}) := \Pr(\gamma(X_1^T, \zeta_1^T) \neq 1 \mid H_1). \tag{1}$$

**Optimization Problem.** Given that human-generated texts can vary widely, within our proposed framework, we aim to control the *worst-case* Type-I error $\sup_{Q_{X_1^T}} \beta_0(\gamma, Q_{X_1^T}, P_{\zeta_1^T})$ at a given $\alpha \in (0,1)$ while minimizing Type-II error. Our objective is to design an $\epsilon$-distorted watermarking scheme and a model-agnostic detector by solving the following optimization:

$$\inf_{\gamma, P_{X_1^T, \zeta_1^T}} \beta_1(\gamma, P_{X_1^T, \zeta_1^T}) \quad \text{s.t.} \sup_{Q_{X_1^T}} \beta_0(\gamma, Q_{X_1^T}, P_{\zeta_1^T}) \leq \alpha, \ \mathsf{D}(P_{X_1^T}, Q_{X_1^T}) \leq \epsilon. \tag{Opt-O}$$

The optimal objective value, denoted as $\beta_1^*(Q_{X_1^T}, \alpha, \epsilon)$, is termed as *universally minimum Type-II error*. This universality is due to its applicability across all potential detectors and watermarking schemes, as well as its validity under the worst-case Type-I error scenario.

## 3 Jointly Optimal Watermarking Scheme and Detector

In this section, we aim to solve the optimization in (Opt-O) and identify the jointly optimal watermarking scheme and detector. However, solving (Opt-O) is challenging due to the binary nature of $\gamma$ and the vast set of possible $\gamma$, sized $2^{|\mathcal{V}|^T |\mathcal{Z}|^T}$. To address this, we begin with a specific $\gamma(X_1^T, \zeta_1^T) = \mathbb{1}\{(X_1^T, \zeta_1^T) \in \mathcal{A}_1\}$, where $\mathcal{A}_1$ defines the acceptance region for $H_1$, aiming to

uncover a potential structure for the optimal detector. To this end, we simplify (Opt-O) as

$$\inf_{P_{X_1^T,\zeta_1^T}} \beta_1(\gamma, P_{X_1^T,\zeta_1^T}) \quad \text{s.t.} \sup_{Q_{X_1^T}} \beta_0(\gamma, Q_{X_1^T}, P_{\zeta_1^T}) \le \alpha,\ \mathsf{D}(P_{X_1^T}, Q_{X_1^T}) \le \epsilon. \tag{Opt-I}$$

**Error-Distortion Tradeoff.** We first derive a lower bound for the minimum Type-II error in (Opt-I), which surprisingly does not depend on the selected detector $\gamma$ and therefore also applies to (Opt-O). We then pinpoint a type of detector and watermarking scheme that attains this lower bound, indicating that it represents the universally minimum Type-II error. Thus, the proposed detector and watermarking scheme are jointly optimal, as detailed in Theorem 2. The theorem below establishes this universal minimum Type-II error for all feasible watermarking schemes and detectors.

**Theorem 1** (Universally minimum Type-II error). *The universally minimum Type-II error attained from* (Opt-O) *is*

$$\beta_1^*(Q_{X_1^T}, \alpha, \epsilon) = \min_{P_{X_1^T}:\mathsf{D}(P_{X_1^T}, Q_{X_1^T}) \le \epsilon} \sum_{x_1^T} (P_{X_1^T}(x_1^T) - \alpha)_+, \tag{2}$$

*which is achieved by the watermarked distribution*

$$P_{X_1^T}^* = \arg\min_{P_{X_1^T}:\mathsf{D}(P_{X_1^T}, Q_{X_1^T}) \le \epsilon} \sum_{x_1^T} (P_{X_1^T}(x_1^T) - \alpha)_+. \tag{3}$$

*By setting* $\mathsf{D}$ *as total variation distance* $\mathsf{D}_{\mathsf{TV}}$, *(2) can be simplified as follows:*

$$\beta_1^*(Q_{X_1^T}, \alpha, \epsilon) = \Big(\sum_{x_1^T} (Q_{X_1^T}(x_1^T) - \alpha)_+ - \epsilon\Big)_+, \quad \text{if } \sum_{x_1^T} (\alpha - Q_{X_1^T}(x_1^T))_+ \ge \epsilon.$$

The proof of Theorem 1 is deferred to Appendix C. Theorem 1 shows that, for any watermarking scheme, the fundamental limits of detection performance depend on the original NTP distribution of the LLM. When the original $Q_{X_1^T}$ is more concentrated (low entropy), the minimum achievable detection error increases. This hints that it is inherently difficult to watermark low-entropy text. However, increasing the allowable distortion $\epsilon$ can enhance the capacity for reducing detection errors, as illustrated in Figure 3. Moreover, we find that $\beta_1^*(Q_{X_1^T}, \alpha, \epsilon)$ matches the minimum Type-II error from Huang et al. [42, Theorem 3.2], which is notably optimal for their specific detector. Our results, however, establish that this is the universally minimum Type-II error across all possible detectors and watermarking schemes, indicating that their detector belongs to the set of optimal detectors described below.

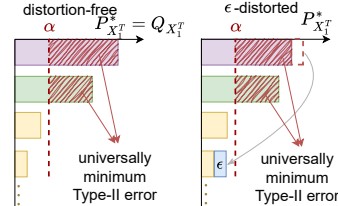

Figure 3: Illustration of error–distortion trade-off.

**Jointly Optimal Design.** We now present the jointly optimal watermarking schemes and detectors that achieve the universally minimum Type-II error in Theorem 1, i.e., the solution to (Opt-O). The key takeaway in the optimal design is: (i) for any given LLM $Q_{X_1^T}$, we can find a valid bijective function $g$ (not unique) to construct a jointly optimal pair of watermarking scheme and detector; (ii) as a result of maximizing the detection performance of dependency against independence between $X_1^T$ and $\zeta_1^T$, the optimal watermarking scheme $P_{\zeta_1^T|X_1^T}^*$ turns out to be a nearly deterministic.

**Theorem 2** ((Informal Statement) Jointly optimal watermarking schemes and detectors). *The class of optimal detectors is given by*

$$\gamma^* \in \Gamma^* \coloneqq \big\{\gamma \mid \gamma(X_1^T, \zeta_1^T) = \mathbb{1}\{X_1^T = g(\zeta_1^T)\}, \text{ for some bijective } g: \mathcal{Z}^T \to \mathcal{S} \supset \mathcal{V}^T\big\}, \tag{4}$$

*where* $\mathcal{Z}^T = g^{-1}(\mathcal{V}^T) \cup \{\tilde\zeta_1^T\}$ *and* $\tilde\zeta_1^T$ *is a redundant auxiliary sequence that is not a preimage of any token sequence for g. Given any* $(Q_{X_1^T}, \epsilon)$, *the corresponding optimal watermarking scheme takes the form* $P_{X_1^T,\zeta_1^T}^* = P_{X_1^T}^* P_{\zeta_1^T|X_1^T}^*$, *where* $P_{X_1^T}^*$ *(c.f. (3)) depends on* $(Q_{X_1^T}, \epsilon)$, *and the mapping* $P_{\zeta_1^T|X_1^T}^*$ *depends on the chosen detector* $\gamma^*$. *Full details are provided in Appendix D.*

In Appendix D, a formal and general statement of Theorem 2 shows that the optimal class of detectors extends to any valid surjective function $g$ with a different input space. To illustrate this with a simple example, suppose $\mathcal{V}^T = \{a, b, c\}$. We can define an extended set $\mathcal{S} = \{a, b, c, \#\}$ and construct a bijective mapping $g$ from an auxiliary set $\mathcal{Z}^T = \{1, 2, 3, 4\}$ to $\mathcal{S}$. Such a class $\Gamma^*$ is then universally optimal, meaning that to guarantee the construction of a watermarking scheme that maximizes the detection performance, the detector must be chosen from $\Gamma^*$.

**Discussions on Theoretically Optimal Watermarking Scheme.** A detailed illustration of the optimal watermarking scheme is provided in Appendix E, with several important remarks as follows.

**First**, we observe that the derived optimal watermarking scheme $P^*_{X_1^T, \zeta_1^T}$ for any $\gamma^* \in \Gamma^*$ is *adaptive* to the original LLM output distribution $Q_{X_1^T}$. This observation suggests that, to maximize watermark detection performance, watermarking schemes should fully leverage generative modeling and make the sampling of auxiliary sequence adaptive to $Q_{X_1^T}$. This approach contrasts with existing watermarking schemes, which typically sample the auxiliary sequence according to a given uniform distribution, without adapting it to the LLM NTP distribution. This insight serves as a foundation for the design of our practical watermarking scheme, which will be introduced in Section 4.

**Second**, in order to control the worst-case FPR, the construction of $P^*_{X_1^T, \zeta_1^T}$ relies on the redundant auxiliary sequence $\tilde{\zeta}_1^T$ included in the auxiliary alphabet $\mathcal{Z}^T$, which satisfies $\gamma^*(x_1^T, \tilde{\zeta}_1^T) = 0$ for all $x_1^T$. This sequence $\tilde{\zeta}_1^T$ plays a critical role in our proposed algorithm. Specifically, if $P^*_{X_1^T}(x_1^T) > \alpha$ (indicating a low-entropy text, e.g., a celebrity's name), it may be mapped to the redundant $\tilde{\zeta}_1^T$, making it harder to detect as watermarked. Thus, the optimal watermarking scheme $P^*_{X_1^T, \zeta_1^T}$ is particularly effective in reducing the FPR for low-entropy texts.

**Practical Challenges.** Given the theoretically optimal structure, there are still a few practical challenges in its direct implementation. ① Designing a proper function $g$, an alphabet $\mathcal{Z}^T$ and the corresponding $P^*_{X_1^T, \zeta_1^T}$ is challenging, as $|\mathcal{V}|^T$ grows exponentially with $T$, making it hard to identify all pairs $(x_1^T, \zeta_1^T)$ such that $x_1^T = g(\zeta_1^T)$. ② The optimality is derived for static scenarios with a fixed token length $T$, making it unsuitable for dynamic scenarios where the tokens are generated incrementally with varying $T$. ③ In the theoretical analysis, we assume full recovery of the auxiliary sequence $\zeta_1^T$ during detection. However, in practice, the detector only receives the token sequence $X_1^T$, and reconstructing the auxiliary sequence $\zeta_1^T$ from $X_1^T$ poses a challenge.

These practical constraints motivate the development of a more feasible version of the theoretically optimal scheme. In Section 4, we adapt it to a practical token-level optimal scheme to address ① and ②; in Section 5, we implement the token-level design with a novel algorithm utilizing a surrogate language model and the Gumbel-Max trick [48] to overcome ③.

## 4 Practical Token-level Optimal Watermarking Design

In this section, we present a practical approach that approximates the theoretical framework while ensuring its applicability to real-world scenarios. Building on the fixed-length optimal scheme, we naturally extend it to accommodate varying-length scenarios by constructing the optimal watermarking scheme incrementally for each token, i.e., $P^*_{X_t, \zeta_t | x_1^{t-1}, \zeta_1^{t-1}}$ for all $t = 1, 2, \ldots$.

To lay the groundwork, we first revisit heuristic detectors for some existing watermarking schemes.

**Example 2** (Examples of heuristic detectors). *Two example detectors from existing works:*
- *Green-Red List watermark detector [25]:* $\gamma(X_1^T, \zeta_1^T) = \mathbb{1}\{\frac{2}{\sqrt{T}}(\sum_{t=1}^T \mathbb{1}\{\zeta_t(X_t) = 1\} - \rho T) \geq \lambda\}$ *where $\lambda > 0$, $\rho \in (0, 1)$, and $\zeta_t = (\zeta_t(x))_{x \in \mathcal{V}}$ is uniformly sampled from $\{\zeta \in \{0, 1\}^{|\mathcal{V}|} : \|\zeta\|_1 = \rho|\mathcal{V}|\}$ with the seed $\text{hash}(X_{t-1}, \text{key})$.*
- *Gumbel-Max watermark detector [38]:* $\gamma(X_1^T, \zeta_1^T) = \mathbb{1}\{-\sum_{t=1}^T \log(1 - \zeta_t(X_t)) \geq \lambda\}\}$ *where $\lambda > 0$, and $\zeta_t = (\zeta_t(x))_{x \in \mathcal{V}}$ is uniformly sampled from $[0, 1]^{|\mathcal{V}|}$ with the seed $\text{hash}(X_{t-1}^{t-n}, \text{key})$ for some $n$.*

**Practical Detector Design.** We observe that the commonly used heuristic detectors take the non-optimal form by averaging the test statistics over each token: $\gamma(X_1^T, \zeta_1^T) = \mathbb{1}\{\frac{1}{T}\sum_{t=1}^T \text{Test Statistics of } (X_t, \zeta_t) \geq \lambda\}$. This token-level design provides several advantages: (1) incremental computation of detectors for any $T$ and 2) token-level watermarking with the alphabet depending only on the fixed size $|\mathcal{V}|$. Inspired by these detectors, we propose the following detector to address the issues ① and ② mentioned earlier:

$$\gamma_{\text{tk}}(X_1^T, \zeta_1^T) = \mathbb{1}\left\{ \frac{1}{T} \sum_{t=1}^{T} \underbrace{\mathbb{1}\{X_t = g_{\text{tk}}(\zeta_t)\}}_{\text{Token-level adaptation of (4)}} \geq \lambda \right\}, \tag{5}$$

for some bijective function $g_{\text{tk}} : \mathcal{Z} \to \mathcal{S} \supset \mathcal{V}$, where $\mathcal{Z} = g_{\text{tk}}^{-1}(\mathcal{V}) \cup \{\tilde{\zeta}\}$ for some redundant auxiliary value $\tilde{\zeta}$ not being the preimage of any token $x \in \mathcal{V}$ for $g$. This detector combines the advantages of existing token-level detectors with the optimal design from Theorem 2. The test statistic for each token $(X_t, \zeta_t)$ is optimal at position $t$, enabling a token-level optimal watermarking scheme that improves the detection performance for each token.

**Token-Level Optimal Watermarking Scheme.** Following the same rule in Theorem 2 and Appendix D, the token-level optimal watermarking scheme is *sequentially* constructed based on $\mathbb{1}\{X_t = g_{\text{tk}}(\zeta_t)\}$ in (5) and the NTP distribution at each position $t$, acting only on the token vocabulary $\mathcal{V}$. This approach addresses the challenges ① and ② as well. Notably, the resulting distribution of the token-level optimal scheme for the auxiliary variable $\zeta_t$ is adaptive to the original NTP distribution $Q_{X_t|x_1^{t-1}}$. Moreover, the resulting distribution on $X_t$ is given by (comparable to $P^*_{X_1^T}$ in Theorem 2)

$$P^*_{X_t|x_1^{t-1}} := \underset{P_{X_t|x_1^{t-1}}:D(P_{X_t|x_1^{t-1}}, Q_{X_t|x_1^{t-1}}) \leq \epsilon}{\arg\min} \sum_{x \in \mathcal{V}} (P_{X_t|x_1^{t-1}}(x) - \eta)_+, \tag{6}$$

where $\eta \in (0,1)$ is the *token-level FPR constraint*, which is typically much greater than the sequence-level FPR constraint $\alpha$. With a proper choice of $\eta$, we can effectively control $\alpha$. Under this scheme, we add watermarks to the generated tokens incrementally, with maximum detection performance at each token. The details are deferred to Appendix F and the algorithm is provided in Section 5.

**Performance Analysis.** We evaluate the Type-I (FPR) and Type-II ($1-$TPR) errors of this scheme over the entire sequence (cf. (1)).

**Lemma 3** ((Informal Statement) Token-level optimal watermarking detection errors)**.** *Under the detector $\gamma_{\text{tk}}$ in (5) and its corresponding token-level optimal watermarking scheme with $\eta \in (0, \min\{1, (\alpha/\binom{T}{\lceil T\lambda \rceil}))^{\frac{1}{\lceil T\lambda \rceil}}\}]$, for a length-$T$ sequence: (i) the worst-case Type-I error $\sup_{Q_{X_1^T}} \beta_0 \leq \alpha$; (ii) if token positions more than $n$ apart are assumed to be independent, with a suitable detector threshold, the Type-II error decays exponentially in $T/n$.*

Although the token-level optimal watermarking scheme may not be optimal at the sequence level, we show that it maintains good performance with a proper choice of token-level FPR $\eta$. The formal statement is provided in Appendix G.

Furthermore, we observe that even without explicitly introducing robustness to the token-level optimal watermarking scheme, it inherently leads to some robustness against token replacement. The following result shows that if the auxiliary sequence $\zeta_1^T$ is shared between the LLM and the detector $\gamma_{\text{tk}}$ (cf. (5)), the token at position $t$ can be replaced with probability $\Pr(\zeta_t \text{ is redundant})$ without affecting detector output.

**Proposition 4** (Robustness against token replacement)**.** *Under the detector $\gamma_{\text{tk}}$ in (5) and its corresponding token-level optimal watermarking scheme, the expected number of tokens that can be randomly replaced in $X_1^T$ without compromising detection performance is $\sum_{t=1}^{T} \mathbb{E}_{X_1^{t-1}} \left[ \sum_{x \in \mathcal{V}} \left( P^*_{X_t|X_1^{t-1}}(x|X_1^{t-1}) - \eta \right)_+ \right]$, with $P^*_{X_t|X_1^{t-1}}$ given in (6).*

## 5    DAWA: Dstribution-Aaptive Watermarking Agorithm

In this section, we implement the token-level design presented in Section 4 by introducing a novel, distortion-free watermarking algorithm, DAWA (**D**istribution-**A**daptive **W**atermarking **A**lgorithm). To address the challenge ③ of recovering the auxiliary sequence at the detector without knowledge of the original LLM and prompt, we utilize some novel tricks, including a surrogate model and Gumbel-Max sampling, which also ensures model-agnosticism and computational efficiency.

**Novel Trick for Auxiliary Sequence Transmission.**    Since the resulting optimal distribution of the auxiliary variable $\zeta_t$ from Section 4 is adaptive to the original NTP distribution of LLM, it is not likely to completely reconstruct it at the detection phase without access to the LLM or prompt. One possible workaround is enforcing $P_{\zeta_t} = \text{Unif}(\mathcal{Z})$ for both watermark generation and detection.

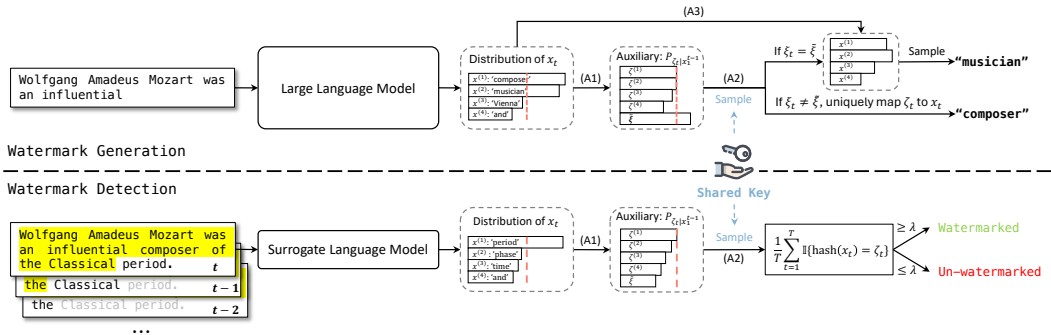

Figure 4: Workflow of our practical algorithm (DAWA) for watermark generation and detection. (A1): construct the sampling distribution of auxiliary variable $\zeta_t$ based on $Q_{x_t|x_1^{t-1},\text{pt}}$; (A2): sample $\zeta_t$ using the Gumbel-Max trick and a shared key; (A3): adjust the NTP distribution of $x_t$ with $\eta$.

While this method simplifies the transmission, it leads to a much higher minimum Type-II error compared to $\beta_1^*(Q_{X_1^T}, \alpha, \epsilon)$ (cf. (2)), indicating a trade-off between detection performance and a non-distribution-adaptive design.

We thus introduce a novel trick to transmit the auxiliary sequence by integrating a *surrogate language model (SLM)* during the detection phase and the Gumbel-Max trick [48] for sampling $\zeta_t$. This SLM, much smaller than the watermarked LLM and possibly from a different family (as long as it shares the same tokenizer) approximates the watermarked distributions $\{P^*_{X_t|X_1^{t-1}}\}_{t=1,2,\ldots}$ using only the text $X_1^T$, without the prompt. With the approximated $P^*_{X_t|X_1^{t-1}}$, we reconstruct the sampling distribution of $\zeta_t$ and sample it using the Gumbel-Max trick with the `key` shared from watermark generation.

Theoretically, the SLM's approximation error has limited impact on detection performance, since the watermarking algorithm is provably resilient to token replacement attacks (cf. Proposition 4). In Section 6, our experiments highlight that, even with *incomplete recovery* of $\zeta_1^T$ during detection, the DAWA algorithm with this novel trick exhibits superior detection performance and greater resilience against token replacement attack, surpassing baseline watermarking schemes.

**DAWA Details.** From the detector in (5), we first choose $g_{\text{tk}}$ that depends on a hash function $h_{\text{key}}$:

$$\gamma_{\text{dawa}}(X_1^T, \zeta_1^T) = \mathbb{1}\Big\{\frac{1}{T} \sum_{t \in [T]} \mathbb{1}\{h_{\text{key}}(X_t) = \zeta_t\} \geq \lambda\Big\}. \tag{7}$$

DAWA is an implementation of the distortion-free ($\epsilon = 0$) token-level optimal watermarking scheme (cf. Section 4), in which the auxiliary variable is sampled adaptively based on the original NTP distribution $Q_{X_t|x_1^{t-1},\text{pt}}$, as illustrated in Figure 4 and detailed in Appendix H. Below, we elaborate on the key steps.

*Watermark Generation.* Using the detector $\gamma_{\text{dawa}}$ from (7), we define the auxiliary alphabet $\mathcal{Z}$ from unique mappings $\{h_{\text{key}}(x)\}_{x \in \mathcal{V}}$ and add a redundant $\tilde{\zeta}$. At each $t$, $P_{\zeta_t|x_1^{t-1},\text{pt}}$ is adaptive to $Q_{X_t|x_1^{t-1},\text{pt}}$:

$$\begin{cases} P_{\zeta_t|x_1^{t-1},\text{pt}}(\zeta) \leftarrow (Q_{X_t|x_1^{t-1},\text{pt}}(h_{\text{key}}^{-1}(\zeta)) \wedge \eta), & \forall \zeta \in \mathcal{Z}\backslash\{\tilde{\zeta}\}. \\ P_{\zeta_t|x_1^{t-1},\text{pt}}(\tilde{\zeta}) \leftarrow \sum_{x \in \mathcal{V}}(Q_{X_t|x_1^{t-1},\text{pt}}(x) - \eta)_+. \end{cases} \tag{A1}$$

The Gumbel-Max trick is then used to sample $\zeta_t$:

$$\zeta_t \leftarrow \arg\max_{\zeta \in \mathcal{Z}} \log(P_{\zeta_t|x_1^{t-1},\text{pt}}(\zeta)) + G_{t,\zeta}. \tag{A2}$$

where $G_{t,\zeta}$ is sampled from the Gumbel distribution using a shared `key` and the previous tokens. If $\zeta_t$ is non-redundant, let $x_t = h_{\text{key}}^{-1}(\zeta_t)$; otherwise, $x_t$ is sampled via a multinomial distribution:

$$x_t \sim \left(\frac{(Q_{X_t|x_1^{t-1},\text{pt}}(x) - \eta)_+}{\sum_{x \in \mathcal{V}}(Q_{X_t|x_1^{t-1},\text{pt}}(x) - \eta)_+}\right)_{x \in \mathcal{V}}. \tag{A3}$$

Table 1: Detection performance on clean and edited text across different LLMs and datasets.

| LLM | Method | Clean Text | | | | | | Token Replacement Attack | | | | | |
| | | C4 | | | ELI5 | | | C4 | | | ELI5 | | |
| | | ROC-AUC | TP@1% FP | TP@10% FP | ROC-AUC | TP@1% FP | TP@10% FP | ROC-AUC | TP@1% FP | TP@10% FP | ROC-AUC | TP@1% FP | TP@10% FP |
|---|---|---|---|---|---|---|---|---|---|---|---|---|---|
| Llama2-13B | KGW+23 | 0.995 | 0.991 | 1.000 | 0.989 | 0.974 | 0.986 | 0.965 | 0.833 | 0.952 | 0.973 | 0.892 | 0.973 |
| | EXP-edit | 0.986 | 0.968 | 0.996 | 0.983 | 0.960 | 0.995 | 0.973 | 0.857 | 0.978 | 0.967 | 0.889 | 0.975 |
| | Gumbel-Max | 0.996 | 0.993 | 0.994 | 0.999 | 0.991 | 0.994 | 0.968 | 0.858 | 0.970 | 0.965 | 0.887 | 0.975 |
| | HCW+23 | 0.994 | 0.928 | 0.986 | 0.991 | 0.888 | 0.978 | 0.890 | 0.268 | 0.714 | 0.893 | 0.278 | 0.692 |
| | **Ours** | 0.999 | 0.998 | 1.000 | 0.998 | 0.997 | 1.000 | 0.989 | 0.860 | 0.976 | 0.995 | 0.969 | 0.994 |
| Mistral-8×7B | KGW+23 | 0.997 | 0.995 | 1.000 | 0.993 | 0.983 | 0.994 | 0.977 | 0.860 | 0.962 | 0.969 | 0.890 | 0.970 |
| | EXP-edit | 0.993 | 0.970 | 0.997 | 0.994 | 0.972 | 0.996 | 0.980 | 0.861 | 0.975 | 0.983 | 0.932 | 0.988 |
| | Gumbel-Max | 0.994 | 0.989 | 0.999 | 0.987 | 0.970 | 0.990 | 0.972 | 0.865 | 0.960 | 0.971 | 0.889 | 0.975 |
| | HCW+23 | 0.998 | 0.986 | 0.994 | 0.999 | 0.992 | 1.000 | 0.885 | 0.364 | 0.674 | 0.878 | 0.252 | 0.668 |
| | **Ours** | 0.999 | 0.998 | 1.000 | 0.999 | 0.999 | 1.000 | 0.990 | 0.881 | 0.966 | 0.993 | 0.991 | 0.995 |

_Watermark Detection._ A surrogate NTP distribution $\tilde{Q}_{X_t|x_1^{t-1}}$ is approximated by the SLM for each $t$. We then use (A1) to approximate $P_{\zeta_t|x_1^{t-1},\text{pt}}$ from $\tilde{Q}_{X_t|x_1^{t-1}}$ and sample $\zeta_t$ using (A2) with the shared key. At each position $t$, the score $\mathbb{1}\{h_{\text{key}}(x_t) = \zeta_t\}$ is 1 if $\zeta_t$ non-redundant and 0 otherwise. Compute $\frac{1}{T}\sum_{t=1}^T \mathbb{1}\{h_{\text{key}}(x_t) = \zeta_t\}$ and compare with a threshold $\lambda$. If above $\lambda$, the text is detected as watermarked.

## 6 Experiments and Discussions

**Experiment Settings.** We now introduce the setup details of our experiments.

_Implementation Details._ Our approach is implemented on two language models: Llama2-13B [1], and Mistral-8×7B [44]. Llama2-7B serves as the surrogate model for Llama2-13B, while Mistral-7B is used as the surrogate model for Mistral-8×7B. We conduct our experiments on Nvidia A100 GPUs. In DAWA, we set $\eta = 0.2$ and $T = 200$.

_Baselines._ We compare our methods with three existing watermarking methods: KGW+23 [25], EXP-edit [37], Gumbel-Max [38], and HCW+23 [19], where the EXP-edit, Gumbel-Max and HCW+23 are distortion-free watermarks. KGW+23 employs the prior 1 token as a hash to create a green/red list, with the watermark strength set at 2.

_Dataset and Prompt._ Our experiments are conducted using two distinct datasets. The first is an open-ended **high-entropy** generation dataset, a realnewslike subset from C4 [49]. The second is a relatively **low-entropy** generation dataset, ELI5 [50]. The realnewslike subset of C4 is tailored specifically to include high-quality journalistic content that mimics

Table 2: Empirical entropy comparison between C4 and ELI5 datasets.

| Model | C4 (entropy) | ELI5 (entropy) |
|---|---|---|
| Llama2-13B | 0.547 | 0.272 |
| Mistral-8×7B | 1.475 | 1.427 |

the style and format of real-world news articles. As shown in Table 2, the C4 dataset consistently exhibits higher empirical entropy than the ELI5 dataset across different models. We utilize the first two sentences of each text as prompts and the following 200 tokens as human-generated text. The ELI5 dataset is specifically designed for the task of long-form question answering (QA), with the goal of providing detailed explanations for complex questions. We use each question as a prompt and its answer as human-generated text.

_Evaluation Metrics._ To evaluate the performance of watermark detection, we report the ROC-AUC score, where the ROC curve shows the True Positive (TP) Rate against the False Positive (FP) Rate. A higher ROC-AUC score indicates better overall performance. The detection threshold $\lambda$ is determined empirically by the ROC-AUC score function based on unwatermarked and watermarked sentences.

### 6.1 Main Experimental Results

**Watermark Detection Performance.** To explore our detection performance at a very low FPR, we conduct experiments using Llama2-13B on $10^5$ texts (200-length) from the Wikipedia dataset and compute the TPR at 1e−01, 1e−02, 1e−03, 1e−04, and 1e−05 FPR respectively. Figure 1 shows that DAWA significantly outperforms other baselines.

Table 3: Comparison of BLEU score and perplexity across different watermarking methods.

| Methods | Human | KGW+23 | EXP-Edit | Gumbel-Max | HCW+23 | **Ours** |
|---|---|---|---|---|---|---|
| BLEU Score ↑ | 0.219 | 0.158 | 0.203 | 0.210 | 0.207 | 0.214 |
| Perplexity ↓ | 8.846 | 13.472 | 10.126 | 9.910 | 10.115 | 10.034 |

Furthermore, we compare the detection performance across various language models and tasks, as presented in Table 1. Our DAWA demonstrates superior performance, especially on the relatively low-entropy QA dataset, validating Theorem 2 and Lemma 3. This success stems from the design of our watermarking scheme, which reduces the likelihood of low-entropy tokens being falsely detected as watermarked, thereby lowering the FPR. Moreover, this suggests that even without knowing the watermarked LLM during detection, we can still use the proposed SLM and Gumbel-Max trick to successfully detect the watermark.

We assess the robustness of DAWA against a **token replacement attack** to validate Proposition 4. For each watermarked text, we randomly mask 50% of the tokens and use T5-large [49] to predict the replacement for each masked token based on the context. Table 1 exhibits watermark detection performance under token replacement attacks across different models and tasks. Our DAWA remains high ROC-AUC, TPR@1%FPR, and TPR@10%FPR under this attack compared with other baselines.

**Watermarked Text Quality.** To evaluate the quality of watermarked text generated by our watermarking methods, we report the perplexity (median) on C4 dataset using GPT-3 [51], and the BLEU score on the machine translation task using the WMT19 dataset [52] and mBART Model [53], as shown in Table 3. It can be observed that our scheme achieves a higher BLEU score and a lower perplexity closer to the unwatermarked one (10.020), both close to the score on human datasets. This demonstrates that our distortion-free scheme, employing an NTP distribution-adaptive approach, has minimal impact on the generated text quality, preserving its naturalness and coherence.

**Ablation Study and Additional Results.** In Appendix I, we further show that (1) our DAWA is efficient and does not affect generation time; (2) detection remains accurate and robust even with a much smaller SLM from a *different model family* and without prompts; (3) TPR increases with longer token length $T$; and (4) our theoretical choice of $\eta$ effectively controls the empirical FPR.

### 6.2 Extension Towards Stronger Robustness

In Appendix J.1, Table 9, we first empirically assess the robustness of DAWA against **random deletion and paraphrasing attacks**. DAWA outperforms Gumbel-Max and KGW+23 in deletion robustness and matches their performance under paraphrasing. These results confirm that DAWA remains competitive while balancing robustness, efficiency, and detection accuracy, with the potential to demonstrate a graceful trade-off based on our theoretical analyses.

**Theoretical Extension.** As a step towards even stronger robustness, Appendix J outlines how our theoretical framework and optimal solutions extend to scenarios involving a wide range of attacks, including *semantic-invariant attacks*. We characterize the detectability–distortion–robustness trade-off and show the closed-form optimal *robust* watermarking scheme–detector pairs. These findings offer valuable insights for designing advanced semantic-based watermarking algorithms that are resilient to such attacks in the future.

## Acknowledgements

This work was performed while Yepeng Liu and Yuheng Bu were with the Department of Electrical and Computer Engineering at the University of Florida. They acknowledge UFIT Research Computing for providing computational resources and support that contributed to the research results reported in this publication. This work was also conducted while Haiyun He was a postdoctoral associate with the Center for Applied Mathematics at Cornell University and while Ziqiao Wang was a Ph.D. student with the School of Electrical Engineering and Computer Science at the University of Ottawa.

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

## A  Other Related Literature

In the past few years, many in-process LLM watermarking methods have been proposed [2, 3, 54–63], including biased and unbiased (distortion-free) ones. Biased watermarks typically alter the next-token prediction (NTP) distribution to increase the likelihood of sampling certain tokens [25–29]. For example, [25] divides the vocabulary into green and red lists and slightly enhances the probability of green tokens in the NTP distribution. Unbiased watermarks maintain the original NTP distributions or texts unchanged, using various sampling strategies to embed watermarks [30–36, 19, 64, 65]. The Gumbel-Max watermark [38] utilizes the Gumbel-Max trick [48] to sample the next token, while Kuditipudi et al. [37] introduces an inverse transform method for this purpose.

From a theoretical standpoint, most of these designs remain heuristic. While post-process watermarking has been extensively studied from an information-theoretic perspective [12–16], the theory behind in-process watermarking is still limited. Prior efforts [42, 43] have analyzed either the watermark embedder or the detector in isolation, without achieving joint or universally optimal designs.

## B  Other Existing Watermarking Schemes

Here, we discuss additional existing watermarking schemes utilizing auxiliary variables, which can be encompassed within our LLM watermarking formulation.

- The **Gumbel-Max watermarking scheme** [38] applies the Gumbel-Max trick [48] to sample the next token $X_t$, where the Gumbel variable is exactly the auxiliary variable $\zeta_t$, which is a $|\mathcal{V}|$-dimensional random vector, indexd by $x$. For $t = 1, 2, \ldots$,

  – Compute a hash using the previous $n$ tokens $X_{t-1}^{t-n}$ and a shared secret key, i.e., $h(X_{t-1}^{t-n}, \text{key})$, where $h : \mathcal{V}^n \times \mathbb{R} \to \mathbb{R}$.
  – Use $h(X_{t-1}^{t-n}, \text{key})$ as a seed to uniformly sample the auxiliary vector $\zeta_t$ from $[0, 1]^{|\mathcal{V}|}$.
  – Sample $X_t$ using the Gumbel-Max trick
  $$X_t = \arg\max_{x \in \mathcal{V}} \log Q_{X_t|x_1^{t-1}}(x) - \log(-\log \zeta_t(x)).$$

- In the **inverse transform watermarking scheme** [37], the vocabulary $\mathcal{V}$ is considered as $[|\mathcal{V}|]$ and the combination of the uniform random variable and the randomly permuted index vector is the auxiliary variable $\zeta_t$.

  – Use key as a seed to uniformly and independently sample $\{U_t\}_{t=1}^T$ from $[0, 1]$, and $\{\pi_t\}_{t=1}^T$ from the space of permutations over $[|\mathcal{V}|]$. Let the auxiliary variable $\zeta_t = (U_t, \pi_t)$, for $t = 1, 2, \ldots, T$.
  – Sample $X_t$ as follows
  $$X_t = \pi_t^{-1}\left( \min\left\{ i \in [|\mathcal{V}|] : \sum_{x \in [|\mathcal{V}|]} \left( Q_{X_t|x_1^{t-1}}(x) \mathbb{1}\{\pi_t(x) \leq i\} \right) \geq U_t \right\} \right),$$
  where $\pi_t^{-1}$ denotes the inverse permutation.

- In **adaptive watermarking** by Liu and Bu [27], the authors introduce a watermarking scheme that adopts a technique similar to the Green-Red List approach but replaces the hash function with a pretrained neural network $h$. The auxiliary variable $\zeta_t$ is sampled from the set $\{\mathbf{v} \in \{0, 1\}^{|\mathcal{V}|} : \|\mathbf{v}\|_1 = \rho|\mathcal{V}|\}$ using the seed $h(\phi(X_1^{t-1}), \text{key})$, where $h$ takes the semantics $\phi(X_1^{t-1})$ of the generated text and the secret key as inputs. They sample $X_t$ using the same process as the Green-Red List approach.

## C  Proof of Theorem 1

According to the Type-I error constraint, we have $\forall x_1^T \in \mathcal{V}^T$,
$$\alpha \geq \max_{Q_{X_1^T}} \mathbb{E}_{Q_{X_1^T} P_{\zeta_1^T}} [\mathbb{1}\{(X_1^T, \zeta_1^T) \in \mathcal{A}_1\}]$$
$$\geq \mathbb{E}_{\delta_{x_1^T} P_{\zeta_1^T}} [\mathbb{1}\{(X_1^T, \zeta_1^T) \in \mathcal{A}_1\}]$$
$$= \mathbb{E}_{P_{\zeta_1^T}} [\mathbb{1}\{(x_1^T, \zeta_1^T) \in \mathcal{A}_1\}]$$

$$= \begin{cases} \sum_{\zeta_1^T} P_{\zeta_1^T}(\zeta_1^T) \mathbb{1}\{(x_1^T, \zeta_1^T) \in \mathcal{A}_1\}, & \mathcal{Z} \text{ is discrete;} \\ \int P_{\zeta_1^T}(\zeta_1^T) \mathbb{1}\{(x_1^T, \zeta_1^T) \in \mathcal{A}_1\} d\zeta_1^T, & \mathcal{Z} \text{ is continuous;} \end{cases}.$$

In the following, for notational simplicity, we assume that $\mathcal{Z}$ is discrete. However, the derivations hold for both discrete $\mathcal{Z}$ and continuous $\mathcal{Z}$. The Type-II error is given by $1 - \mathbb{E}_{P_{X_1^T,\zeta_1^T}}[\mathbb{1}\{(X_1^T, \zeta_1^T) \in \mathcal{A}_1\}]$. We have

$$\mathbb{E}_{P_{X_1^T,\zeta_1^T}}[\mathbb{1}\{(X_1^T, \zeta_1^T) \in \mathcal{A}_1\}] = \sum_{x_1^T} \underbrace{\sum_{\zeta_1^T} P_{X_1^T,\zeta_1^T}(x_1^T, \zeta_1^T) \mathbb{1}\{(x_1^T, \zeta_1^T) \in \mathcal{A}_1\}}_{C(x_1^T)},$$

where for all $x_1^T \in \mathcal{V}^T$,

$$C(x_1^T) \le P_{X_1^T}(x_1^T) \quad \text{and} \quad C(x_1^T) \le \sum_{\zeta_1^T} P_{\zeta_1^T}(\zeta_1^T) \mathbb{1}\{(x_1^T, \zeta_1^T) \in \mathcal{A}_1\} \le \alpha$$

according to the Type-I error bound. Therefore,

$$\mathbb{E}_{P_{X_1^T,\zeta_1^T}}[\mathbb{1}\{(X_1^T, \zeta_1^T) \in \mathcal{A}_1\}] = \sum_{x_1^T} C(x_1^T) \le \sum_{x_1^T}(P_{X_1^T}(x_1^T) \wedge \alpha)$$

$$= 1 - \sum_{x_1^T}(P_{X_1^T}(x_1^T) - \alpha)_+ \tag{8}$$

where (8) is maximized at

$$P_{X_1^T}^* := \underset{P_{X_1^T}:D(P_{X_1^T},Q_{X_1^T}) \le \epsilon}{\arg\min} \sum_{x_1^T}(P_{X_1^T}(x_1^T) - \alpha)_+. \tag{9}$$

For any $P_{X_1^T}$, the Type-II error is lower bounded by

$$\mathbb{E}_{P_{X_1^T,\zeta_1^T}}[\mathbb{1}\{(X_1^T, \zeta_1^T) \notin \mathcal{A}_1\}] \ge \sum_{x_1^T}(P_{X_1^T}(x_1^T) - \alpha)_+.$$

By plugging $P_{X_1^T}^*$ into this lower bound, we obtain a Type-II lower bound that holds for all $\gamma$ and $P_{X_1^T,\zeta_1^T}$. Recall that Huang et al. [42] proposed a type of detector and watermarking scheme that achieved this lower bound. As we demonstrate, it is actually the universal minimum Type-II error over all possible $\gamma$ and $P_{X_1^T,\zeta_1^T}$, denoted by $\beta_1^*(Q_{X_1^T}, \epsilon, \alpha)$.

Specifically, define $\epsilon^*(x_1^T) = Q_{X_1^T}(x_1^T) - P_{X_1^T}^*(x_1^T)$ and we have

$$\sum_{x_1^T:P_{X_1^T}^*(x_1^T) \ge \alpha} \epsilon^*(x_1^T) = \sum_{x_1^T:P_{X_1^T}^*(x_1^T) \ge \alpha, \epsilon^*(x_1^T) \ge 0} \epsilon^*(x_1^T) + \underbrace{\sum_{x_1^T:P_{X_1^T}^*(x_1^T) \ge \alpha, \epsilon^*(x_1^T) \le 0} \epsilon^*(x_1^T)}_{\le 0}$$

$$\le \sum_{x_1^T:P_{X_1^T}^*(x_1^T) \ge \alpha, \epsilon^*(x_1^T) \ge 0} \epsilon^*(x_1^T)$$

$$= \sum_{x_1^T:P_{X_1^T}^*(x_1^T) \ge \alpha, Q_{X_1^T}(x_1^T) \ge P_{X_1^T}^*(x_1^T)} \epsilon^*(x_1^T)$$

$$\le \sum_{x_1^T:Q_{X_1^T}(x_1^T) \ge P_{X_1^T}^*(x_1^T)} \epsilon^*(x_1^T) \le \epsilon$$

where the last inequality follows from the total variation distance constraint $\mathsf{D}_{\mathsf{TV}}(P_{X_1^T}, Q_{X_1^T}) \le \epsilon$. We rewrite $\beta_1^*(Q_{X_1^T}, \epsilon, \alpha)$ as follows:

$$\beta_1^*(Q_{X_1^T}, \epsilon, \alpha) = \underset{P_{X_1^T}:\mathsf{D}_{\mathsf{TV}}(P_{X_1^T},Q_{X_1^T}) \le \epsilon}{\min} \sum_{x_1^T}(P_{X_1^T}(x_1^T) - \alpha)_+$$

$$= \sum_{x_1^T:P_{X_1^T}^*(x_1^T) \ge \alpha}(P_{X_1^T}^*(x_1^T) - \alpha),$$

$$= \sum_{x_1^T:P_{X_1^T}^*(x_1^T) \ge \alpha}(Q_{X_1^T}(x_1^T) - \epsilon^*(x_1^T) - \alpha)$$

$$= \sum_{x_1^T : P^*_{X_1^T}(x_1^T) \geq \alpha} (Q_{X_1^T}(x_1^T) - \alpha) - \sum_{x_1^T : P^*_{X_1^T}(x_1^T) \geq \alpha} \epsilon^*(x_1^T)$$

$$\geq \sum_{x_1^T} (Q_{X_1^T}(x_1^T) - \alpha)_+ - \epsilon,$$

where the last inequality follows from $\sum_{x_1^T : P^*_{X_1^T}(x_1^T) \geq \alpha} \epsilon^*(x_1^T) \leq \epsilon$, i.e. the total variation constraint limits how much the distribution $P^*_{X_1^T}$ can be perturbed from $Q_{X_1^T}$. Since $\beta_1^*(Q_{X_1^T}, \epsilon, \alpha) \geq 0$, finally we have

$$\beta_1^*(Q_{X_1^T}, \epsilon, \alpha) \geq \left( \sum_{x_1^T} (Q_{X_1^T}(x_1^T) - \alpha)_+ - \epsilon \right)_+ .$$

Notably, the lower bound is achieved when $\{x_1^T : P^*_{X_1^T}(x_1^T) \geq \alpha\} = \{x_1^T : Q_{X_1^T}(x_1^T) \geq P^*_{X_1^T}(x_1^T)\}$ and $D_{TV}(Q_{X_1^T}, P^*_{X_1^T}) = \epsilon$. That is, to construct $P^*_{X_1^T}$, an $\epsilon$ amount of the mass of $Q_{X_1^T}$ above $\alpha$ is moved to below $\alpha$, which is possible only when $\sum_{x_1^T} (\alpha - Q_{X_1^T}(x_1^T))_+ \geq \epsilon$. Note that Huang et al. [42, Theorem 3.2] points out a sufficient condition for this to hold: $|\mathcal{V}|^T \geq \frac{1}{\alpha}$. The optimal distribution $P^*_{X_1^T}$ thus satisfies

$$\sum_{x_1^T : Q_{X_1^T}(x_1^T) \geq \alpha} (Q_{X_1^T}(x_1^T) - P^*_{X_1^T}(x_1^T)) = \sum_{x_1^T : Q_{X_1^T}(x_1^T) \leq \alpha} (P^*_{X_1^T}(x_1^T) - Q_{X_1^T}(x_1^T)) = \epsilon.$$

**Refined constraints for optimization.** We notice that the feasible region of (Opt-I) can be further reduced as follows:

$$\min_{P_{X_1^T}} \min_{P_{\zeta_1^T | X_1^T}} \quad \mathbb{E}_{P_{X_1^T} P_{\zeta_1^T | X_1^T}} [1 - \gamma(X_1^T, \zeta_1^T)] \qquad \text{(Opt-II)}$$

$$\text{s.t.} \quad \int P_{\zeta_1^T | X_1^T}(\zeta_1^T | x_1^T) \, d\zeta_1^T = 1, \; \forall x_1^T$$

$$\int P_{\zeta_1^T | X_1^T}(\zeta_1^T | x_1^T) \gamma(x_1^T, \zeta_1^T) \leq 1 \wedge \frac{\alpha}{P_{X_1^T}(x_1^T)}, \; \forall x_1^T \qquad (10)$$

$$D_{TV}(P_{X_1^T}, Q_{X_1^T}) \leq \epsilon,$$

$$\sup_{Q_{X_1^T}} \sum_{x_1^T} Q_{X_1^T}(x_1^T) \int \left( \sum_{y_1^T} P_{\zeta_1^T | X_1^T}(\zeta_1^T | y_1^T) P_{X_1^T}(y_1^T) \right) \gamma(x_1^T, \zeta_1^T) \, d\zeta_1^T \leq \alpha,$$

where (10) is an additional constraint on $P_{\zeta_1^T | X_1^T}$. If and only if (10) can be achieved with equality, the minimum of the objective function $\mathbb{E}_{P_{X_1^T} P_{\zeta_1^T | X_1^T}} [1 - \gamma(X_1^T, \zeta_1^T)]$ reaches (2).

# D   Formal Statement of Theorem 2 and its Proof

**Theorem 2 [Formal]** (Optimal type of detectors and watermarking schemes)**.** *The set of all detectors that achieve the minimum Type-II error $\beta_1^*(Q_{X_1^T}, \alpha, \epsilon)$ in Theorem 1 for all text distribution $Q_{X_1^T} \in \mathcal{P}(\mathcal{V}^T)$ and distortion level $\epsilon \geq 0$ is precisely*
$$\Gamma^* := \left\{ \gamma \,|\, \gamma(X_1^T, \zeta_1^T) = \mathbb{1}\{X_1^T = g(\zeta_1^T)\}, \text{ for some surjective } g : \mathcal{Z}^T \to \mathcal{S} \supset \mathcal{V}^T \right\}.$$
*For any valid function $g$, choose a redundant auxiliary value $\tilde{\zeta}_1^T \in \mathcal{Z}^T$ such that $x_1^T \neq g(\tilde{\zeta}_1^T)$ for all $x_1^T \in \mathcal{V}^T$. The detailed construction of the optimal watermarking scheme is as follows:*
$$P^*_{X_1^T} = \min_{P_{X_1^T} : D(P_{X_1^T}, Q_{X_1^T}) \leq \epsilon} \sum_{x_1^T} (P_{X_1^T}(x_1^T) - \alpha)_+,$$

*and for any $x_1^T \in \mathcal{V}^T, P^*_{\zeta_1^T | X_1^T}(\zeta_1^T | x_1^T)$ satisfies* $\qquad (11)$

$$\begin{cases} P^*_{X_1^T}(x_1^T) \sum_{\zeta_1^T} P^*_{\zeta_1^T | X_1^T}(\zeta_1^T | x_1^T) \gamma(x_1^T, \zeta_1^T) = P^*_{X_1^T}(x_1^T) \wedge \alpha, & \forall \zeta_1^T \text{ s.t. } \gamma(x_1^T, \zeta_1^T) = 1; \\ P^*_{X_1^T}(x_1^T) P^*_{\zeta_1^T | X_1^T}(\zeta_1^T | x_1^T) = (P^*_{X_1^T}(x_1^T) - \alpha)_+, & \text{if } \zeta_1^T = \tilde{\zeta}_1^T; \\ P^*_{\zeta_1^T | X_1^T}(\zeta_1^T | x_1^T) = 0, & \text{otherwise.} \end{cases}$$

*Proof.* First, we observe that the lower bound on the Type-II error in (2) is attained if and only if the constraint in (10) holds with equality for all $x_1^T$ and for the optimizer. Thus, it suffices to show that for any detector $\gamma \notin \Gamma^*$, the constraint in (10) cannot hold with equality for all $x_1^T$ given any text distributions $Q_{X_1^T}$. First, define an arbitrary surjective function $g : \mathcal{Z}^T \to \mathcal{S}$, where $\mathcal{S}$ is on the same metric space as $\mathcal{V}^T$. Cases 1 and 2 prove that $\mathcal{V}^T \subset \mathcal{S}$. Case 3 proves that $\gamma$ can only be $\gamma(X_1^T, \zeta_1^T) = \mathbb{1}\{X_1^T = g(\zeta_1^T)\}$.

- **Case 1:** $\gamma(X_1^T, \zeta_1^T) = \mathbb{1}\{X_1^T = g(\zeta_1^T)\}$ but $\mathcal{S} \subset \mathcal{V}^T$. There exists $\tilde{x}_1^T$ such that for all $\zeta_1^T$, $\mathbb{1}\{\tilde{x}_1^T = g(\zeta_1^T)\} = 0$. Under this case, (10) cannot hold with equality for $\tilde{x}_1^T$ since the LHS is always 0 while the RHS is positive.

- **Case 2:** $\gamma(X_1^T, \zeta_1^T) = \mathbb{1}\{X_1^T = g(\zeta_1^T)\}$ but $\mathcal{S} = \mathcal{V}^T$. Let us start from the simple case where $T = 1$, $\mathcal{V} = \{x_1, x_2\}$, $\mathcal{Z} = \{\zeta_1, \zeta_2\}$, and $g$ is an identity mapping. Given any $Q_X$ and any feasible $P_X$ such that $\mathsf{D}_{\mathsf{TV}}(P_X, Q_X) \le \epsilon$, when (10) holds with equality, i.e.,
  $$P_{X,\zeta}(x_1, \zeta_1) = P_X(x_1) \wedge \alpha \quad \text{and} \quad P_{X,\zeta}(x_2, \zeta_2) = P_X(x_2) \wedge \alpha,$$
  then the marginal $P_\zeta$ is given by: $P_\zeta(\zeta_1) = P_X(x_1) \wedge \alpha + (P_X(x_2) - \alpha)_+$, $P_\zeta(\zeta_2) = P_X(x_2) \wedge \alpha + (P_X(x_1) - \alpha)_+$. The worst-case Type-I error is given by
  $$\sup_{Q_X} \left( Q_X(x_1)\big(P_X(x_1) \wedge \alpha + (P_X(x_2) - \alpha)_+\big) + Q_X(x_2)\big(P_X(x_2) \wedge \alpha + (P_X(x_1) - \alpha)_+\big) \right)$$

  $$\ge P_X(x_1) \wedge \alpha + (P_X(x_2) - \alpha)_+$$
  $$> \alpha, \quad \text{if } P_X(x_1) > \alpha, P_X(x_2) > \alpha.$$
  It implies that for any $Q_X$ such that $\{P_X \in \mathcal{P}(\mathcal{V}) : \mathsf{D}_{\mathsf{TV}}(P_X, Q_X) \le \epsilon\} \subseteq \{P_X \in \mathcal{P}(\mathcal{V}) : P_X(x_1) > \alpha, P_X(x_2) > \alpha\}$, the false-alarm constraint is violated when (10) holds with equality. It can be verified that this result also holds for larger $(T, \mathcal{V}, \mathcal{Z})$ and other functions $g : \mathcal{Z}^T \to \mathcal{V}^T$.

- **Case 3:** Let $\Xi_\gamma(x_1^T) := \{\zeta_1^T \in \mathcal{Z}^T : \gamma(x_1^T, \zeta_1^T) = 1\}$. $\exists x_1^T \ne y_1^T \in \mathcal{V}^T$, s.t. $\Xi(x_1^T) \cap \Xi(y_1^T) \ne \emptyset$. For any detector $\gamma \notin \Gamma^*$ that does not fall into Cases 1 and 2, it falls into Case 3. Let us start from the simple case where $T = 1$, $\mathcal{V} = \{x_1, x_2\}$, $\mathcal{Z} = \{\zeta_1, \zeta_2, \zeta_3\}$. Consider a detector $\gamma$ as follows: $\gamma(x_1, \zeta_1) = \gamma(x_2, \zeta_1) = 1$ and $\gamma(x, \zeta) = 0$ for all other pairs $(x, \zeta) \in \mathcal{V} \times \mathcal{Z}$. Hence, $\Xi(x_1) \cap \Xi(x_2) = \{\zeta_1\}$. When (10) holds with equality, i.e.,
  $$P_{X,\zeta}(x_1, \zeta_1) = P_X(x_1) \wedge \alpha \quad \text{and} \quad P_{X,\zeta}(x_2, \zeta_1) = P_X(x_2) \wedge \alpha,$$
  we have the worst-case Type-I error lower bounded by
  $$\sup_{Q_X} \left( Q_X(x_1)P_\zeta(\zeta_1) + Q_X(x_2)P_\zeta(\zeta_1) \right) = P_\zeta(\zeta_1) = P_X(x_1) \wedge \alpha + P_X(x_2) \wedge \alpha$$

  $$> \alpha, \quad \text{if } P_X(x_1) > \alpha \text{ or } P_X(x_2) > \alpha.$$
  Thus, for any $Q_X$ such that $\{P_X \in \mathcal{P}(\mathcal{V}) : \mathsf{D}_{\mathsf{TV}}(P_X, Q_X) \le \epsilon\} \subseteq \{P_X \in \mathcal{P}(\mathcal{V}) : P_X(x_1) > \alpha \text{ or } P_X(x_2) > \alpha\}$, the false-alarm constraint is violated when (10) holds with equality.

  If we consider a detector $\gamma$ as follows: $\gamma(x_1, \zeta_1) = \gamma(x_2, \zeta_1) = \gamma(x_2, \zeta_2) = 1$ and $\gamma(x, \zeta) = 0$ for all other pairs $(x, \zeta) \in \mathcal{V} \times \mathcal{Z}$. We still have $\Xi(x_1) \cap \Xi(x_2) = \{\zeta_1\}$. When (10) holds with equality, i.e.,
  $$P_{X,\zeta}(x_1, \zeta_1) = P_X(x_1) \wedge \alpha \quad \text{and} \quad P_{X,\zeta}(x_2, \zeta_1) + P_{X,\zeta}(x_2, \zeta_2) = P_X(x_2) \wedge \alpha,$$
  we have the worst-case Type-I error lower bounded by
  $$\sup_{Q_X} \left( Q_X(x_1)P_\zeta(\zeta_1) + Q_X(x_2)(P_\zeta(\zeta_1) + P_\zeta(\zeta_2)) \right) = \sup_{Q_X} \left( P_\zeta(\zeta_1) + Q_X(x_2)P_\zeta(\zeta_2) \right)$$
  $$= P_\zeta(\zeta_1) + P_\zeta(\zeta_2) = P_X(x_1) \wedge \alpha + P_X(x_2) \wedge \alpha > \alpha, \quad \text{if } P_X(x_1) > \alpha \text{ or } P_X(x_2) > \alpha,$$
  which is the same as the previous result.

  If we let $\mathcal{V} = \{x_1, x_2, x_3\}$, $\mathcal{Z} = \{\zeta_1, \zeta_2, \zeta_3, \zeta_4\}$ and $\gamma(x_3, \zeta_3) = 1$ in addition to the afore-mentioned $\gamma$, we can similarly show that the worst-case Type-I error is larger than $\alpha$ for some distributions $Q_X$.

  Therefore, it can be observed that as long as $\Xi(x_1^T) \cap \Xi(y_1^T) \ne \emptyset$ for some $x_1^T \ne y_1^T \in \mathcal{V}^T$, (10) can not be achieved with equality for all $Q_{X_1^T}$ and $\epsilon$ even for larger $(T, \mathcal{V}, \mathcal{Z})$ as well as continuous $\mathcal{Z}$.

In conclusion, for any detector $\gamma \notin \Gamma^*$, the universal minimum Type-II error in (2) cannot be obtained for all $Q_{X_1^T}$ and $\epsilon$.

Since the optimal detector takes the form $\gamma(X_1^T, \zeta_1^T) = \mathbb{1}\{X_1^T = g(\zeta_1^T)\}$ for some surjective function $g : \mathcal{Z}^T \to \mathcal{S}, \mathcal{S} \supset \mathcal{V}^T$, and the token vocabulary is discrete, it suffices to consider discrete $\mathcal{Z}$ to derive the optimal watermarking scheme.

Under the watermarking scheme $P^*_{X_1^T, \zeta_1^T}$ (cf. (9) and (11)), the Type-I and Type-II errors are given by:

**Type-I error:**
$$\forall y_1^T \in \mathcal{V}^T, \quad \mathbb{E}_{P^*_{\zeta_1^T}}[\mathbb{1}\{y_1^T = g(\zeta_1^T)\}] = \sum_{\zeta_1^T} P^*_{\zeta_1^T}(\zeta_1^T)\mathbb{1}\{y_1^T = g(\zeta_1^T)\}$$

$$= \sum_{\zeta_1^T} \sum_{x_1^T} P^*_{X_1^T, \zeta_1^T}(x_1^T, \zeta_1^T)\mathbb{1}\{y_1^T = g(\zeta_1^T)\}$$

$$= P^*_{X_1^T}(y_1^T) \sum_{\zeta_1^T} P^*_{\zeta_1^T|X_1^T}(\zeta_1^T|y_1^T)\mathbb{1}\{y_1^T = g(\zeta_1^T)\} = P^*_{X_1^T}(y_1^T) \wedge \alpha$$

$$\leq \alpha,$$

and since any distribution $Q_{X_1^T}$ can be written as a linear combinations of $\delta_{y_1^T}$, we have

$$\max_{Q_{X_1^T}} \mathbb{E}_{Q_{X_1^T} P^*_{\zeta_1^T}}[\mathbb{1}\{X_1^T = g(\zeta_1^T)\}] \leq \alpha.$$

**Type-II error:**
$$1 - \mathbb{E}_{P^*_{X_1^T, \zeta_1^T}}[\mathbb{1}\{X_1^T = g(\zeta_1^T)\}]$$

$$= 1 - \sum_{x_1^T} \sum_{\zeta_1^T} P^*_{X_1^T, \zeta_1^T}(x_1^T, \zeta_1^T)\mathbb{1}\{x_1^T = g(\zeta_1^T)\}$$

$$= 1 - \sum_{x_1^T} P^*_{X_1^T}(x_1^T) \sum_{\zeta_1^T} P^*_{\zeta_1^T|X_1^T}(\zeta_1^T|x_1^T)\mathbb{1}\{x_1^T = g(\zeta_1^T)\}$$

$$= 1 - \sum_{x_1^T} \left(P^*_{X_1^T}(x_1^T) \wedge \alpha\right)$$

$$= \sum_{x_1^T : P^*_{X_1^T}(x_1^T) > \alpha} (P^*_{X_1^T}(x_1^T) - \alpha).$$

The optimality of $P^*_{X_1^T, \zeta_1^T}$ is thus proved. We note that (10) in (Opt-II) holds with equality under this optimal conditional distribution $P^*_{\zeta_1^T|X_1^T}$.

Compared to Huang et al. [42, Theorem 3.2], their proposed detector is equivalent to $\gamma(X_1^T, \zeta_1^T) = \mathbb{1}\{X_1^T = \zeta_1^T\}$, where $\mathcal{Z}^T = \mathcal{V}^T \cup \{\tilde{\zeta}_1^T\}$ and $\tilde{\zeta}_1^T \notin \mathcal{V}^T$, meaning that it belongs to $\Gamma^*$. □

# E   Illustration of Construction of the Optimal Watermarking Scheme

Using a toy example in Figure 5, we now illustrate how to construct the optimal watermarking schemes, where
$$P^*_{X_1^T} = \arg\min_{P_{X_1^T} : D(P_{X_1^T}, Q_{X_1^T}) \leq \epsilon} \sum_{x_1^T} (P_{X_1^T}(x_1^T) - \alpha)_+.$$

Constructing the optimal watermarking scheme $P^*_{X_1^T, \zeta_1^T}$ is equivalent to transporting the probability mass $P^*_{X_1^T}$ on $\mathcal{V}$ to $\mathcal{Z}$, maximizing $P^*_{X_1^T, \zeta_1^T}(x_1^T, \zeta_1^T)$ when $x_1^T = g(\zeta_1^T)$, while keeping the worst-case Type-I error below $\alpha$. Without loss of generality, by letting $T = 1$, we present Figure 5 to visualize the optimal watermarking scheme. The construction process is given step by step as follows:
– **Identify text-auxiliary pairs:** We begin by identifying text-auxiliary pairs $(x, \zeta) \in \mathcal{V} \times \mathcal{Z}$ with $\gamma(x, \zeta) = \mathbb{1}\{x = g(\zeta)\} = 1$ and connect them by blue solid lines.

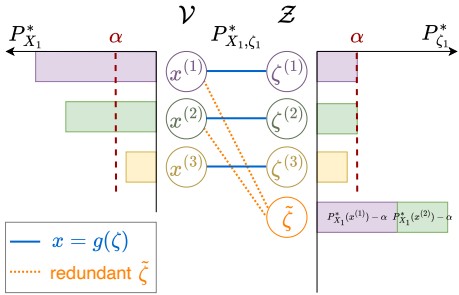

Figure 5: A toy example of the optimal detector and watermarking scheme when $T = 1$. Links between $\mathcal{V}$ and $\mathcal{Z}$ suggest $P^*_{X_1, \zeta_1} > 0$.

– **Introducing redundant auxiliary value:** We enlarge $\mathcal{Z}$ to include an additional value $\tilde{\zeta}$ and set $\gamma(x, \tilde{\zeta}) = 0$ for all $x$. We will call $\tilde{\zeta}$ "redundant".

– **Mass allocation for $P^*_{X_1}(x) > \alpha$:** If $P^*_{X_1}(x) > \alpha$, we transfer $\alpha$ mass of $P^*_{X_1}(x)$ to the $\zeta$ connected by the blue solid lines. The excess mass is transferred to the redundant $\tilde{\zeta}$ (orange dashed lines). Specifically, for $x^{(1)}$, where $P^*_{X_1}(x^{(1)}) > \alpha$ and $x^{(1)} = g(\zeta^{(1)})$, we move $\alpha$ units of mass from $P^*_{X_1}(x^{(1)})$ to $P^*_{\zeta_1}(\zeta^{(1)})$, ensuring that $P^*_{\zeta_1}(\zeta^{(1)}) = \alpha$. The rest $(P^*_{X_1}(x^{(1)}) - \alpha)$ units of mass is moved to $\tilde{\zeta}$. Similarly, for $x^{(2)}$, where $P^*_{X_1}(x^{(2)}) > \alpha$ and $x^{(2)} = g(\zeta^{(2)})$, we move $\alpha$ mass from $P^*_{X_1}(x^{(2)})$ to $P^*_{\zeta_1}(\zeta^{(2)})$ and $(P^*_{X_1}(x^{(2)}) - \alpha)$ mass to $\tilde{\zeta}$. Consequently, the probability of $\tilde{\zeta}$ is $P_{\zeta_1}(\tilde{\zeta}) = (P^*_{X_1}(x^{(1)}) - \alpha) + (P^*_{X_1}(x^{(2)}) - \alpha)$. In this way, there is a chance for the lower-entropy texts $x^{(1)}$ and $x^{(2)}$ to be mapped to the redundant $\tilde{\zeta}$ during watermark generation.

– **Mass allocation for $P^*_{X_1}(x) < \alpha$:** For $x^{(3)}$, where $P^*_{X_1}(x^{(3)}) < \alpha$ and $x^{(3)} = g(\zeta^{(3)})$, we move the entire mass $P^*_{X_1}(x^{(3)})$ to $P^*_{\zeta_1}(\zeta^{(3)})$ along the blue solid line. It means that higher-entropy texts will not be mapped to the redundant $\tilde{\zeta}$ during watermark generation.

– **Outcome:** This construction ensures that $P^*_{\zeta_1}(\zeta) \le \alpha$ for all $\zeta \in \{\zeta^{(1)}, \zeta^{(2)}, \zeta^{(3)}\}$, keeping the worst-case Type-I error under control. The Type-II error is equal to $P^*_{\zeta_1}(\tilde{\zeta})$, which is exactly the universally minimum Type-II error. This scheme can be similarly generalized to $T > 1$.

In Figure 5, when there is no link between $(x, \zeta) \in \mathcal{V} \times \mathcal{Z}$, the joint probability $P^*_{X_1, \zeta_1}(x, \zeta) = 0$. By letting $\epsilon = 0$, the scheme guarantees that the watermarked LLM remains unbiased (distortion-free). Note that the detector proposed in Huang et al. [42, Theorem 3.2] is also included in our framework, see Appendix D.

## F  Construction of Token-level Optimal Watermarking Scheme

The toke-level optimal watermarking scheme is the optimal solution to the following optimization problem:

$$\inf_{P_{X_t, \zeta_t | X_1^{t-1}, \zeta_1^{t-1}}} \mathbb{E}_{P_{X_t, \zeta_t | X_1^{t-1}, \zeta_1^{t-1}}} [1 - \mathbb{1}\{X_t = g_{\mathrm{tk}}(\zeta_t)\}]$$

$$\text{s.t.} \sup_{Q_{X_t | X_1^{t-1}}} \mathbb{E}_{Q_{X_t | X_1^{t-1}} \otimes P_{\zeta_t | \zeta_1^{t-1}}} [\mathbb{1}\{X_t = g_{\mathrm{tk}}(\zeta_t)\}] \le \eta, \ \mathsf{D}_{\mathsf{TV}}(P_{X_t | X_1^{t-1}}, Q_{X_t | X_1^{t-1}}) \le \epsilon.$$

The optimal solution $P^*_{X_t, \zeta_t | X_1^{t-1}, \zeta_1^{t-1}}$ follows the similar rule as that of $P^*_{X_1^T, \zeta^T}$ in Theorem 2 with $(Q_{X_1^T}, P_{X_1^T}, \alpha)$ replaced by $(Q_{X_t | X_1^{t-1}}, P_{X_t | X_1^{t-1}}, \eta)$. We refer readers to Appendix D for further details.

## G  Formal Statement of Lemma 3 and its Proof

Let $P^{\mathrm{token}*}_{X_1^T, \zeta_1^T}$ and $P^{\mathrm{token}*}_{\zeta_1^T}$ denote the joint distributions induced by the token-level optimal watermarking scheme.

**Lemma 3 (Formal)** (Token-level optimal watermarking detection errors). *Let $\eta = (\alpha/(\binom{T}{\lceil T\lambda \rceil}))^{\frac{1}{\lceil T\lambda \rceil}}$.*
*Under the detector $\gamma$ in (5) and the token-level optimal watermarking scheme $P^*_{X_t,\zeta_t|X_1^{t-1},\zeta_1^{t-1}}$, the*
*Type-I error is upper bounded by*

$$\sup_{Q_{X_1^T}} \beta_0(\gamma, Q_{X_1^T}, P_{\zeta_1^T}^{\text{token}*}) \leq \alpha.$$

*Assume that when $T$ and $n \leq T$ are both large enough, token $X_t$ is independent of $X_{t-i}$, i.e.,*
*$P_{X_t, X_{t-i}} = P_{X_t} \otimes P_{X_{t-i}}$, for all $i \geq n+1$ and $t \in [T]$. Let $\mathcal{I}_{T,n}(i) = ([i-n, i+n] \cap [T])\backslash\{i\}$.*
*By setting the detector threshold as $\lambda = \frac{a}{T} \sum_{t=1}^T \mathbb{E}_{X_t,\zeta_t}[\mathbb{1}\{X_t = g(\zeta_t)\}]$ for some $a \in [0,1]$, the*
*Type-II error exponent is*

$$-\log \beta_1(\gamma, P_{X_1^T,\zeta_1^T}^{\text{token}*}) = \Omega\left(\frac{T}{n}\right).$$

The following is the proof of Lemma 3.

To choose $\lceil T\lambda \rceil$ indices out of $\{1, \ldots, T\}$, there are $\binom{T}{\lceil T\lambda \rceil}$ choices. Let $k = 1, \ldots, \binom{T}{\lceil T\lambda \rceil}$ and $S_k$ be the $k$-th set of the chosen indices. The Type-I error is upper-bounded by

$$\beta_0(\gamma, Q_{X^{(T)}}, P_{\zeta_1^T}^{\text{token}*}) = \Pr\left(\frac{1}{T}\sum_{t=1}^T \mathbb{1}\{X_t = g(\zeta_t)\} \geq \lambda \mid H_0\right)$$

$$\leq \Pr\left(\bigcup_{k=1}^{\binom{T}{\lceil T\lambda \rceil}} \{\mathbb{1}\{X_t = g(\zeta_t)\} = 1, \forall t \in S_k\} \mid H_0\right)$$

$$\leq \sum_{k=1}^{\binom{T}{\lceil T\lambda \rceil}} \underbrace{\Pr\left(\{\mathbb{1}\{X_t = g(\zeta_t)\} = 1, \forall t \in S_k\} \mid H_0\right)}_{P_{\text{FA},k}}.$$

Without loss of generality, let $m = \lceil T\lambda \rceil$ and $S_k = \{1, 2, \ldots, m\}$. We can rewrite $P_{\text{FA},k}$ as

$$P_{\text{FA},k} = \mathbb{E}_{Q_{X^{(T)}} \otimes P_{\zeta^{(T)}}}[\{\mathbb{1}\{X_t = g(\zeta_t)\} = 1, \forall t \in S_k\}]$$

$$= \mathbb{E}_{Q_{X^{(T)}} \otimes P_{\zeta^{(T)}}}[\prod_{t \in S_k} \mathbb{1}\{X_t = g(\zeta_t)\}]$$

$$= \mathbb{E}_{Q_{X_1} \otimes P_{\zeta_1}}\left[\mathbb{1}\{X_1 = g(\zeta_1)\}\mathbb{E}_{Q_{X_2|X_1} \otimes P_{\zeta_2|\zeta_1}}\left[\mathbb{1}\{X_2 = g(\zeta_2)\} \cdots\right.\right.$$

$$\left.\left.\cdots \mathbb{E}_{Q_{X_m|X_1^{m-1}} \otimes P_{\zeta_m|\zeta_1^{m-1}}}[\mathbb{1}\{X_m = g(\zeta_m)\}]\right] \cdots\right]$$

$$\leq \eta^m, \quad \forall Q_{X_1^T}.$$

Then the Type-I error is finally upper-bounded by

$$\sup_{Q_{X_1^T}} \beta_0(\gamma, Q_{X_1^T}, P_{\zeta_1^T}^{\text{token}*}) \leq \binom{T}{\lceil T\lambda \rceil}\eta^{\lceil T\lambda \rceil} \leq \alpha.$$

We prove the Type-II error bound by applying Janson [66, Theorem 10].

**Theorem 5** (Theorem 10, Janson [66]). *Let $\{I_i\}_{i \in \mathcal{I}}$ be a finite family of indicator random variables, defined on a common probability space. Let $G$ be a dependency graph of $\mathcal{I}$, i.e., a graph with vertex set $\mathcal{I}$ such that if $A$ and $B$ are disjoint subsets of $\mathcal{I}$, and $\Gamma$ contains no edge between $A$ and $B$, then $\{I_i\}_{i \in A}$ and $\{I_i\}_{i \in B}$ are independent. We write $i \sim j$ if $i, j \in \mathcal{I}$ and $(i,j)$ is an edge in $G$. In particular, $i \nsim i$. Let $S = \sum_{i \in \mathcal{I}} I_i$ and $\Delta = \mathbb{E}[S]$. Let $\Psi = \max_{i \in \mathcal{I}} \sum_{j \in \mathcal{I}, j \sim i} \mathbb{E}[I_j]$ and $\Phi = \frac{1}{2}\sum_{i \in \mathcal{I}}\sum_{j \in \mathcal{I}, j \sim i} \mathbb{E}[I_i I_j]$. For any $0 \leq a \leq 1$,*

$$\Pr(S \leq a\Delta) \leq \exp\left\{-\min\left\{(1-a)^2 \frac{\Delta^2}{8\Phi + 2\Delta}, (1-a)\frac{\Delta}{6\Psi}\right\}\right\}. \tag{12}$$

Given any detector $\gamma$ that accepts the form in (5) and the corresponding optimal watermarking scheme, for some $a \in (0,1)$, we first set the threshold in $\gamma$ as

$$T\lambda = a \sum_{t=1}^{T} \mathbb{E}_{X_t,\zeta_t}[\mathbb{1}\{X_t = g(\zeta_t)\}] = a \sum_{t=1}^{T} \mathbb{E}_{X_1^{t-1}}\left[ \sum_x \left(P^*_{X_t|X_1^{t-1}}(x|X_1^{t-1}) - \eta\right)_+ \right] =: a\Delta_T,$$

where $P^*_{X_t|X_1^{t-1}}$ is induced by $P^*_{X_t,\zeta_t|X_1^{t-1},\zeta_1^{t-1}}$. The Type-II error is given by

$$\beta_1(\gamma, P^{\text{token}*}_{X_1^T,\zeta_1^T}) = P^{\text{token}*}_{X_1^T,\zeta_1^T}\left( \sum_{t=1}^{T} \mathbb{1}\{X_t = g(\zeta_t)\} < a\Delta_T \right)$$

which is exactly the left-hand side of (12).

Assume that when $T$ and $n \leq T$ are large enough, token $X_t$ is independent of all $X_{t-i}$ for all $i \geq n+1$ and $t \in [T]$, i.e., $P_{X_t, X_{t-i}} = P_{X_t} \otimes P_{X_{t-i}}$. Let $\mathcal{I}_{T,n}(i) = ([i-n, i+n] \cap [T]) \backslash \{i\}$. The $\Psi$ and $\Phi$ on the right-hand side of (12) are given by:

$$\Psi := \max_{i \in [T]} \sum_{t \in [T], t \sim i} \mathbb{E}_{X_t,\zeta_t}[\mathbb{1}\{X_t = g(\zeta_t)\}] = \max_{i \in [T]} \sum_{t \in \mathcal{I}_{T,n}(i)} \mathbb{E}_{X_t,\zeta_t}[\mathbb{1}\{X_t = g(\zeta_t)\}] = \Theta(n),$$

$$\Phi := \frac{1}{2} \sum_{i \in [T]} \sum_{j \in [T], j \sim i} \mathbb{E}[\mathbb{1}\{X_i = g(\zeta_i)\}\mathbb{1}\{X_j = g(\zeta_j)\}]$$

$$= \frac{1}{2} \sum_{i \in [T]} \sum_{j \in \mathcal{I}_{T,n}(i)} \mathbb{E}[\mathbb{1}\{X_i = g(\zeta_i)\}\mathbb{1}\{X_j = g(\zeta_j)\}] = \Theta(Tn).$$

By plugging $\Delta_T$, $\Omega$ and $\Theta$ back into the right-hand side of (12), we have the upper bound

$$\beta_1(\gamma, P^{\text{token}*}_{X_1^T,\zeta_1^T}) \leq \exp\left\{ -\min\left\{ (1-a)^2 \frac{\Delta_T^2}{8\Phi + 2\Delta_T}, (1-a)\frac{\Delta_T}{6\Psi} \right\} \right\}$$

where $U_t = \mathbb{E}_{X_1^{t-1}}\left[ \sum_x \left(P^*_{X_t|X_1^{t-1}}(x|X_1^{t-1}) - \eta\right)_+ \right]$, $\Delta_T := \sum_{t=1}^{T} U_t$, $\Psi = \max_{i \in [T]} \sum_{t \in \mathcal{I}_{T,n}(i)} U_t$, and $\Phi = \frac{1}{2} \sum_{i \in [T]} \sum_{j \in \mathcal{I}_{T,n}(i)} \mathbb{E}[\mathbb{1}\{X_i = g(\zeta_i)\}\mathbb{1}\{X_j = g(\zeta_j)\}]$. This implies

$$-\log \beta_1(\gamma, P^{\text{token}*}_{X_1^T,\zeta_1^T}) \geq \min\left\{ (1-a)^2 \Theta\left(\frac{T}{n}\right), (1-a)\Theta\left(\frac{T}{n}\right) \right\}$$

$$\implies -\log \beta_1(\gamma, P^{\text{token}*}_{X_1^T,\zeta_1^T}) = \Omega\left(\frac{T}{n}\right).$$

## H  DAWA Pseudo-Codes

---

**Algorithm 1** Watermarked Text Generation

---

**Input:** LLM $Q$, Vocabulary $\mathcal{V}$, Prompt $u$, Secret key, Token-level false alarm $\eta$.

1: $\mathcal{Z} = \{h_{\text{key}}(x)\}_{x \in \mathcal{V}} \cup \{\tilde{\zeta}\}$

2: **for** $t = 1, ..., T$ **do**

3:     $P_{\zeta_t|x_1^{t-1},u}(\zeta) \leftarrow (Q_{X_t|x_1^{t-1},u}(h_{\text{key}}^{-1}(\zeta)) \wedge \eta), \forall \zeta \in \mathcal{Z} \backslash \{\tilde{\zeta}\}.$

4:     $P_{\zeta_t|x_1^{t-1},u}(\tilde{\zeta}) \leftarrow \sum_{x \in \mathcal{V}}(Q_{X_t|x_1^{t-1},u}(x) - \eta)_+.$

5:     Compute a hash of tokens $x_{t-n}^{t-1}$ with key, and use it as a seed to generate $(G_{t,\zeta})_{\zeta \in \mathcal{Z}}$ from Gumbel distribution.

6:     $\zeta_t \leftarrow \arg\max_{\zeta \in \mathcal{Z}} \log(P_{\zeta_t|x_1^{t-1},u}(\zeta)) + G_{t,\zeta}.$

7:     **if** $\zeta_t \neq \tilde{\zeta}$ **then**

8:       $x_t \leftarrow h_{\text{key}}^{-1}(\zeta_t)$

9:     **else**

10:       Sample $x_t \sim \left( \frac{(Q_{X_t|x_1^{t-1},u}(x)-\eta)_+}{\sum_{x \in \mathcal{V}} \left(Q_{X_t|x_1^{t-1},u}(x)-\eta\right)_+} \right)_{x \in \mathcal{V}}$

11:     **end if**

12: **end for**

**Output:** Watermarked text $x_1^T = (x_1, ..., x_T)$.

---

**Algorithm 2** Watermarked Text Detection

**Input:** SLM $\tilde{Q}$, Vocabulary $\mathcal{V}$, Text $x_1^T$, Secret key, Token-level false alarm $\eta$, Threshold $\lambda$.

1: score $= 0$, $\mathcal{Z} = \{h_{\text{key}}(x)\}_{x \in \mathcal{V}} \cup \{\tilde{\zeta}\}$
2: **for** $t = 1, ..., T$ **do**
3: $\quad \tilde{P}_{\zeta_t | x_1^{t-1}}(\zeta) \leftarrow (\tilde{Q}_{X_t | x_1^{t-1}}(h_{\text{key}}^{-1}(\zeta)) \wedge \eta), \; \forall \zeta \in \mathcal{Z} \backslash \{\tilde{\zeta}\}.$
4: $\quad \tilde{P}_{\zeta_t | x_1^{t-1}}(\tilde{\zeta}) \leftarrow \sum_{x \in \mathcal{V}} (\tilde{Q}_{X_t | x_1^{t-1}}(x) - \eta)_+.$
5: $\quad$ Compute a hash of tokens $x_{t-n}^{t-1}$ with key, and use it as a seed to generate $(G_{t,\zeta})_{\zeta \in \mathcal{Z}}$ from Gumbel distribution.
6: $\quad \zeta_t \leftarrow \arg\max_{\zeta \in \mathcal{Z}} \log(\tilde{P}_{\zeta_t | x_1^{t-1}}(\zeta)) + G_{t,\zeta}.$
7: $\quad$ score $\leftarrow$ score $+ \mathbb{1}\{h_{\text{key}}(x_t) = \zeta_t\}$
8: **end for**
9: **if** score $> T\lambda$ **then**
10: $\quad$ **return** 1 $\qquad\qquad\qquad$ ▷ Input text is watermarked
11: **else**
12: $\quad$ **return** 0 $\qquad\qquad\qquad$ ▷ Input text is unwatermarked
13: **end if**

# I   Ablation Study and Additional Experimental Results

All pre-existing models and datasets utilized in this research are publicly available and were used in full accordance with their respective licensing terms, which predominantly include common open-source licenses (such as Apache 2.0, MIT, CC BY-SA) and specific community or research usage agreements.

**Efficiency of Watermark Scheme.**   To evaluate the efficiency of our watermarking method, we conduct experiments to measure the average generation time for both watermarked and unwatermarked text. In both scenarios, we generated $500$ texts, each containing $200$ tokens. Table 4 indicates that the difference in generation time between unwatermarked and watermarked text is less than $0.5$ seconds. This minimal difference confirms that our watermarking method has a negligible impact on generation speed, ensuring practical applicability.

Table 4: Average generation time comparison for watermarked and unwatermarked text using Llama2-13B.

| Language Model | Setting | Avg Generation Time (s) |
|---|---|---|
| Llama2-13B | Unwatermarked | 9.110 |
| Llama2-13B | Watermarked | 9.386 |

Table 5: Performance comparison of different language models and surrogate models under two scenarios: without attack and with token replacement attack.

| Scenario | Language Model | Surrogate Model | ROC-AUC | TPR@1% FPR | TPR@10% FPR |
|---|---|---|---|---|---|
| Without Attack | Llama2-13B | Llama2-7B | 0.999 | 0.998 | 1.000 |
| | Mistral-8 × 7B | Mistral-7B | 0.999 | 0.998 | 1.000 |
| | GPT-J-6B | GPT-2 large | 0.997 | 0.990 | 0.997 |
| With Attack | Llama2-13B | Llama2-7B | 0.989 | 0.860 | 0.976 |
| | Mistral-8 × 7B | Mistral-7B | 0.990 | 0.881 | 0.966 |
| | GPT-J-6B | GPT-2 large | 0.987 | 0.892 | 0.962 |

**Surrogate Language Model.**   SLM plays a crucial role during the detection process of our watermarking method. We examine how the choice of SLM affects the detection performance of our watermarking scheme. The selection of the surrogate model is primarily based on its vocabulary or tokenizer rather than the specific language model within the same family. This choice is critical because, during detection, the text must be tokenized exactly using the same tokenizer as the watermarking model to ensure accurate token recovery. As a result, any language model that employs the

same tokenizer can function effectively as the surrogate model. To validate our approach, we apply our watermarking algorithm to GPT-J-6B (a model with 6 billion parameters) and use GPT-2 Large (774 million parameters) as the SLM. Despite differences in developers, training data, architecture, and training methods, these two models share the same tokenizer, making them compatible for this task. We conduct experiments using the C4 dataset, and the results are presented in Table 5. The results demonstrate the effectiveness of our proposed watermarking method with or without attack, even when using a surrogate model from a different family than the watermarking language model. Notably, the surrogate model, despite having fewer parameters and lower overall capability compared to the watermarking language model, does not compromise the watermarking performance.

**Prompt Agnostic.** Prompt agnosticism is a crucial property of LLM watermark detection. We investigate the impact of prompts on our watermark detection performance by conducting experiments to compare detection accuracy with and without prompts attached to the watermarked text during the detection process. The results are presented in Table 6. Notably, even when prompts are absent and the SLM cannot perfectly reconstruct the same distribution of $\zeta_t$ as in the generation process, our detection performance remains almost unaffected. This demonstrates the robustness of our watermarking method, regardless of whether a prompt is included during the detection phase.

Table 6: Performance comparison of Llama2-13B under two scenarios: without attack and with token replacement attack, with and without prompts.

| Scenario | Language Model | Surrogate Model | Setting | ROC-AUC | TPR@1% FPR | TPR@10% FPR |
|---|---|---|---|---|---|---|
| Without Attack | Llama2-13B | Llama2-7B | Without Prompt | 0.997 | 0.983 | 0.995 |
| | Llama2-13B | Llama2-7B | With Prompt | 0.998 | 0.989 | 0.996 |
| With Attack | Llama2-13B | Llama2-7B | Without Prompt | 0.977 | 0.818 | 0.953 |
| | Llama2-13B | Llama2-7B | With Prompt | 0.979 | 0.816 | 0.960 |

**Detection Performance with larger $T$.** Increasing text length generally improves detection performance for LLM watermarking. We conduct an additional experiment with $T = 500$, and the results are shown below. Both DAWA and KGW+23 show improved performance compared to $T = 200$ (reported in Figure 1). Notably, DAWA, a distortion-free algorithm, achieves significantly better detection in the low-FPR regime than the distorted KGW+23.

Table 7: Detection performance at various FPRs for different sequence lengths $T$.

| Length $T$ | Method | TPR@1e-5FPR | TPR@1e-4FPR | TPR@1e-3FPR | TPR@1e-2FPR |
|---|---|---|---|---|---|
| 500 | KGW+23 | 0.876 | 0.959 | 0.986 | 0.995 |
| | Ours | **0.891** | **0.970** | **0.996** | **0.999** |
| 200 | KGW+23 | 0.682 | 0.916 | 0.976 | 0.991 |
| | Ours | **0.882** | **0.951** | **0.992** | **0.997** |

**Empirical analysis on False Alarm Control.** We conduct experiments to show the relationship between theoretical FPR (i.e., $\alpha$) and the corresponding empirical FPR. As discussed in Lemma 3, we set the token-level false alarm rate as $\eta = 0.1$ and the sequence length as $T = 200$, which controls the sequence-level false alarm rate under $\alpha = \binom{T}{\lceil T\lambda \rceil} \eta^{\lceil T\lambda \rceil}$, where $\lambda$ is the detection threshold. For a given theoretical FPR $\alpha$, we calculate the corresponding threshold $\lambda$ and the empirical FPR based on 100k unwatermarked sentences. The results, as shown in Table 8, confirm that our theoretical guarantee effectively controls the empirical false alarm rate.

Table 8: Theoretical and empirical FPR under different thresholds.

| Theoretical FPR | 9e-03 | 2e-03 | 5e-04 | 9e-05 |
|---|---|---|---|---|
| Empirical FPR | 1e-04 | 4e-05 | 2e-05 | 2e-05 |

## J  Theoretical Extension to Robustness against Broader Attacks

Thus far, we have theoretically examined the optimal detector and watermarking scheme without considering adversarial scenarios. In practice, users may attempt to modify LLM output to remove watermarks through techniques like replacement, deletion, insertion, paraphrasing, or translation. We now show that our framework can be extended to incorporate robustness against these attacks.

### J.1  Assessment of DAWA against Deletion and Paraphrasing Attacks

We conducted additional experiments on random deletion attacks and paraphrasing attacks, where DAWA achieves comparable robustness to Gumbel-Max and KGW+23. Although it is less robust than EXP-edit, we note that EXP-edit explicitly includes robustness designs and is significantly slower in detection. This demonstrates that DAWA remains competitive while balancing robustness, efficiency, and detection accuracy, with the potential to demonstrate a graceful tradeoff based on our theoretical analyses.

Table 9: Detection performance under deletion and paraphrasing attacks.

| Method | Deletion Attacks | | | Paraphrasing Attacks | | |
|---|---|---|---|---|---|---|
| | ROC-AUC | TPR@1%FPR | TPR@10%FPR | ROC-AUC | TPR@1%FPR | TPR@10%FPR |
| KGW+23 | 0.895 | 0.523 | 0.809 | 0.769 | 0.156 | 0.455 |
| Gumbel-Max | 0.910 | 0.501 | 0.823 | 0.773 | 0.152 | 0.463 |
| EXP-edit | 0.978 | 0.955 | 0.970 | 0.853 | 0.245 | 0.703 |
| DAWA (Ours) | 0.918 | 0.504 | 0.812 | 0.770 | 0.144 | 0.458 |

### J.2  Theoretical Analysis of $f$-Robust Design

We consider a broad class of attacks, where the text can be altered in arbitrary ways as long as certain latent pattern, such as its *semantics*, is preserved. Specifically, let $f : \mathcal{V}^T \to [K]$ be a function that maps a sequence of tokens $X_1^T$ to a finite latent space $[K] \subset \mathbb{N}_+$; for example, $[K]$ may index $K$ distinct semantics clusters and $f$ is a function extracting the semantics. Clearly, $f$ induces an equivalence relation, say, denoted by $\equiv_f$, on $\mathcal{V}^T$, where $x_1^T \equiv_f x'^T_1$ if and only if $f(x_1^T) = f(x'^T_1)$. Let $\mathcal{B}_f(x_1^T)$ be an equivalence class containing $x_1^T$. Under the assumption that the adversary is arbitrarily powerful except that it is unable to move any $x_1^T$ outside its equivalent class $\mathcal{B}_f(x_1^T)$ (e.g., unable to alter the semantics of $x_1^T$), the "$f$-robust" Type-I and Type-II errors are then defined as

$$\beta_0(\gamma, Q_{X_1^T}, P_{\zeta_1^T}, f) := \mathbb{E}_{Q_{X_1^T} \otimes P_{\zeta_1^T}} \left[ \sup_{\tilde{x}_1^T \in \mathcal{B}_f(X_1^T)} \mathbb{1}\{\gamma(\tilde{x}_1^T, \zeta_1^T) = 1\} \right],$$

$$\beta_1(\gamma, P_{X_1^T, \zeta_1^T}, f) := \mathbb{E}_{P_{X_1^T, \zeta_1^T}} \left[ \sup_{\tilde{x}_1^T \in \mathcal{B}_f(X_1^T)} \mathbb{1}\{\gamma(\tilde{x}_1^T, \zeta_1^T) = 0\} \right].$$

Designing a universally optimal $f$-robust detector and watermarking scheme can then be formulated as jointly minimizing the $f$-robust Type-II error while constraining the worst-case $f$-robust Type-I error, namely, solving the optimization problem

$$\inf_{\gamma, P_{X_1^T, \zeta_1^T}} \beta_1(\gamma, P_{X_1^T, \zeta_1^T}, f) \quad \text{s.t.} \sup_{Q_{X_1^T}} \beta_0(\gamma, Q_{X_1^T}, P_{\zeta_1^T}, f) \leq \alpha, \ \mathsf{D}_{\mathsf{TV}}(P_{X_1^T}, Q_{X_1^T}) \leq \epsilon. \quad \text{(Opt-R)}$$

We prove the following theorem.

**Theorem 6** (Universally minimum $f$-robust Type-II error). *The universally minimum $f$-robust Type-II error attained from* (Opt-R) *is*

$$\beta_1^*(Q_{X_1^T}, \alpha, \epsilon, f) := \min_{P_{X_1^T} : \mathsf{D}(P_{X_1^T}, Q_{X_1^T}) \leq \epsilon} \sum_{k \in [K]} \left( \left( \sum_{x_1^T : f(x_1^T) = k} P_{X_1^T}(x_1^T) \right) - \alpha \right)_+.$$

Notably, $\beta_1^*(Q_{X_1^T}, \alpha, \epsilon, f)$ is suboptimal without an adversary but becomes optimal under the adversarial setting of (Opt-R). The gap between $\beta_1^*(Q_{X_1^T}, \alpha, \epsilon, f)$ in Theorem 6 and $\beta_1^*(Q_{X_1^T}, \alpha, \epsilon)$ in Theorem 1 reflects the cost of ensuring robustness, widening as $K$ decreases (i.e., as perturbation strength increases), see Figure 6 in appendix for an illustration of the optimal $f$-robust minimum Type-II error when $f$ is a semantic mapping. Similar to Theorem 2, we derive the optimal detector and watermarking scheme achieving $\beta_1^*(Q_{X_1^T}, \alpha, \epsilon, f)$, detailed in Appendix L. These solutions closely resemble those in Theorem 2. For implementation, if the latent space $[K]$ is significantly

smaller than $\mathcal{V}^T$, applying the optimal $f$-robust detector and watermarking scheme becomes more effective than those presented in Theorem 2. Additionally, a similar algorithmic strategy to the one discussed in Sections 4 and 5 can be employed to address the practical challenges discussed earlier. These extensions and efficient implementations of the function $f$ in practice are promising directions of future research.

## K   Proof of Theorem 6

According to the Type-I error constraint, we have $\forall x_1^T \in \mathcal{V}^T$,

$$
\alpha \geq \max_{Q_{X_1^T}} \mathbb{E}_{Q_{X_1^T} \otimes P_{\zeta_1^T}} \left[ \sup_{\tilde{x}_1^T \in \mathcal{B}_f(X_1^T)} \mathbb{1}\{\gamma(\tilde{x}_1^T, \zeta_1^T) = 1\} \right]
$$

$$
\geq \mathbb{E}_{\delta_{x_1^T} \otimes P_{\zeta_1^T}} \left[ \sup_{\tilde{x}_1^T \in \mathcal{B}_f(X^T)} \mathbb{1}\{\gamma(\tilde{x}_1^T, \zeta_1^T) = 1\} \right] = \mathbb{E}_{P_{\zeta_1^T}} \left[ \sup_{\tilde{x}_1^T \in \mathcal{B}_f(x_1^T)} \gamma(\tilde{x}_1^T, \zeta_1^T) \right]
$$

$$
= \sum_{\zeta_1^T} P_{\zeta_1^T}(\zeta_1^T) \sup_{\tilde{x}_1^T \in \mathcal{B}_f(x_1^T)} \gamma(\tilde{x}_1^T, \zeta_1^T).
$$

For brevity, let $\mathcal{B}(k) := \mathcal{B}_f(x_1^T)$ if $f(x_1^T) = k$. The $f$-robust Type-II error is equal to $1 - \mathbb{E}_{P_{X_1^T, \zeta_1^T}}[\inf_{\tilde{x}_1^T \in \mathcal{B}_f(X_1^T)} \gamma(\tilde{x}_1^T, \zeta_1^T)]$. We have

$$
\mathbb{E}_{P_{X_1^T, \zeta_1^T}} \left[ \inf_{\tilde{x}_1^T \in \mathcal{B}_f(X_1^T)} \gamma(\tilde{x}_1^T, \zeta_1^T) \right] \leq \mathbb{E}_{P_{X_1^T, \zeta_1^T}} \left[ \sup_{\tilde{x}_1^T \in \mathcal{B}_f(X_1^T)} \gamma(\tilde{x}_1^T, \zeta_1^T) \right]
$$

$$
= \sum_{k \in [K]} \underbrace{\sum_{x_1^T : f(x_1^T)=k} \sum_{\zeta_1^T} P_{X_1^T, \zeta_1^T}(x_1^T, \zeta_1^T) \sup_{\tilde{x}_1^T \in \mathcal{B}_f(x_1^T)} \gamma(\tilde{x}_1^T, \zeta_1^T)}_{C(k)},
$$

where according to the $f$-robust Type-I error constraint, for all $k \in [K]$,

$$
C(k) \leq \sum_{x_1^T : f(x_1^T)=k} P_{X_1^T}(x_1^T), \quad \text{and}
$$

$$
C(k) = \sum_{\zeta_1^T} P_{\zeta_1^T}(\zeta_1^T) \sum_{x_1^T : f(x_1^T)=k} P_{X_1^T | \zeta_1^T}(x_1^T | \zeta_1^T) \sup_{\tilde{x}_1^T \in \mathcal{B}(k)} \gamma(\tilde{x}_1^T, \zeta_1^T)
$$

$$
\leq \sum_{\zeta_1^T} P_{\zeta_1^T}(\zeta_1^T) \sup_{\tilde{x}_1^T \in \mathcal{B}(k)} \gamma(\tilde{x}_1^T, \zeta_1^T) \leq \alpha.
$$

Therefore,

$$
\mathbb{E}_{P_{X_1^T, \zeta_1^T}} \left[ \inf_{\tilde{x}_1^T \in \mathcal{B}(f(X_1^T))} \gamma(\tilde{x}_1^T, \zeta_1^T) \right] \leq \sum_{k \in [K]} C(k)
$$

$$
\leq \sum_{k \in [K]} \left( \left( \sum_{x_1^T : f(x_1^T)=k} P_{X_1^T}(x_1^T) \right) \wedge \alpha \right) = 1 - \sum_{k \in [K]} \left( \left( \sum_{x_1^T : f(x_1^T)=k} P_{X_1^T}(x_1^T) \right) - \alpha \right)_+ \quad (13)
$$

where (13) is maximized by taking

$$
P_{X_1^T} = P_{X_1^T}^{*,f} := \operatorname*{arg\,min}_{P_{X_1^T} : D(P_{X_1^T}, Q_{X_1^T}) \leq \epsilon} \sum_{k \in [K]} \left( \left( \sum_{x_1^T : f(x_1^T)=k} P_{X_1^T}(x_1^T) \right) - \alpha \right)_+.
$$

For any $P_{X_1^T}$, the $f$-robust Type-II error is lower bounded by

$$
\mathbb{E}_{P_{X_1^T, \zeta_1^T}} \left[ \sup_{\tilde{x}_1^T \in \mathcal{B}_f(X_1^T)} \mathbb{1}\{\gamma(\tilde{x}_1^T, \zeta_1^T) = 0\} \right] \geq \sum_{k \in [K]} \left( \left( \sum_{x_1^T : f(x_1^T)=k} P_{X_1^T}(x_1^T) \right) - \alpha \right)_+.
$$

By plugging $P_{X_1^T}^{*,f}$ into the lower bound, we obtain the universal minimum $f$-robust Type-II error over all possible $\gamma$ and $P_{X_1^T, \zeta_1^T}$, denoted by

$$
\beta_1^*(f, Q_{X_1^T}, \epsilon, \alpha) := \min_{P_{X_1^T} : D(P_{X_1^T}, Q_{X_1^T}) \leq \epsilon} \sum_{k \in [K]} \left( \left( \sum_{x_1^T : f(x_1^T)=k} P_{X_1^T}(x_1^T) \right) - \alpha \right)_+. \quad (14)
$$

## L   Optimal Type of $f$-Robust Detectors and Watermarking Schemes

**Theorem 7** (Optimal type of $f$-robust detectors and watermarking schemes). *Let $\Gamma_f^*$ be a collection of detectors that accept the form*
$$\gamma(X_1^T, \zeta_1^T) = \mathbb{1}\{X_1^T = g(\zeta_1^T) \text{ or } f(X_1^T) = g(\zeta_1^T)\}$$
*for some function $g : \mathcal{Z}^T \to \mathcal{S}$, $\mathcal{S} \cap ([K] \cup \mathcal{V}^T) \neq \emptyset$ and $|\mathcal{S}| > K$. If and only if the detector $\gamma \in \Gamma_f^*$, the minimum Type-II error attained from* (Opt-R) *reaches $\beta_1^*(Q_{X_1^T}, \epsilon, \alpha, f)$ in* (14) *for all text distribution $Q_{X_1^T} \in \mathcal{P}(\mathcal{V}^T)$ and distortion level $\epsilon \in \mathbb{R}_{\geq 0}$.*

*After enlarging $\mathcal{Z}^T$ to include redundant auxiliary values, the $\epsilon$-distorted optimal $f$-robust watermarking scheme $P_{X_1^T, \zeta_1^T}^{*,f}(x_1^T, \zeta_1^T)$ is given as follows:*

$$P_{X_1^T}^{*,f} := \underset{P_{X_1^T} : \mathsf{D}_{\mathsf{TV}}(P_{X_1^T}, Q_{X_1^T}) \leq \epsilon}{\arg\min} \sum_{k \in [K]} \left( \left( \sum_{x_1^T : f(x_1^T) = k} P_{X_1^T}(x_1^T) \right) - \alpha \right)_+,$$

*and for any $x_1^T \in \mathcal{V}^T$,*

*1) for all $\zeta_1^T$ s.t. $\sup_{\tilde{x}_1^T \in \mathcal{B}(f(x_1^T))} \gamma(\tilde{x}_1^T, \zeta_1^T) = 1$: $P_{\zeta_1^T | X_1^T}^{*,f}(\zeta_1^T | x_1^T)$ satisfies*

$$\sum_{\tilde{x}_1^T \in \mathcal{B}_f(x_1^T)} P_{X_1^T}^{*,f}(\tilde{x}_1^T) \sum_{\zeta_1^T} P_{\zeta_1^T | X_1^T}^{*,f}(\zeta_1^T | \tilde{x}_1^T) \sup_{\tilde{x}_1^T \in \mathcal{B}_f(x_1^T)} \gamma(\tilde{x}_1^T, \zeta_1^T) = \left( \sum_{\tilde{x}_1^T \in \mathcal{B}_f(x_1^T)} P_{X_1^T}^{*,f}(\tilde{x}_1^T) \right) \wedge \alpha.$$

*2) $\forall \zeta_1^T$ s.t. $|\{x_1^T \in \mathcal{V}^T : \gamma(x_1^T, \zeta_1^T) = 1\}| = 0$: $P_{X_1^T, \zeta_1^T}^{*,f}(x_1^T, \zeta_1^T)$ satisfies*

$$\sum_{\tilde{x}_1^T \in \mathcal{B}_f(x_1^T)} P_{X_1^T}^{*,f}(x_1^T) \sum_{\zeta_1^T : |\{x_1^T : \gamma(x_1^T, \zeta_1^T) = 1\}| = 0} P_{\zeta_1^T | X_1^T}^{*,f}(\zeta_1^T | x_1^T) = \left( \left( \sum_{\tilde{x}_1^T \in \mathcal{B}_f(x_1^T)} P_{X_1^T}^{*,f}(\tilde{x}_1^T) \right) - \alpha \right)_+.$$

*3) all other cases of $\zeta_1^T$: $P_{X_1^T, \zeta_1^T}^{*,f}(x_1^T, \zeta_1^T) = 0$.*

*Proof of Theorem 7.* When $f$ is an identity mapping, it is equivalent to Theorem 2. When $f : \mathcal{V}^T \to [K]$ is some other function, following from the proof of Theorem 2, we consider three cases.

- **Case 1:** $\mathcal{S} \cap ([K] \cup \mathcal{V}^T) \neq \emptyset$ but $|\mathcal{S}| < K$. It is impossible for the detector to detect all the watermarked text sequences. That is, there exist $\tilde{x}_1^T$ such that for all $\zeta_1^T$, $\gamma(\tilde{x}_1^T, \zeta_1^T) = 0$. Under this case, in Appendix K, $C(f(\tilde{x}_1^T)) = 0 \neq (\sum_{x_1^T : f(x_1^T) = f(\tilde{x}_1^T)} P_{X_1^T}(x_1^T)) \wedge \alpha$, which means the $f$-robust Type-II error cannot reach the lower bound.

- **Case 2:** $\mathcal{S} \cap ([K] \cup \mathcal{V}^T) \neq \emptyset$ but $|\mathcal{S}| = K$. Under this condition, the detector needs to accept the form $\gamma(X_1^T, \zeta_1^T) = \mathbb{1}\{f(X_1^T) = g(\zeta_1^T)\}$ so as to detect all possible watermarked text. Otherwise, it will degenerate to Case 1. We can see $f(X_1^T)$ as an input variable and rewrite the detector as $\gamma'(f(X_1^T), \zeta_1^T) = \gamma(X_1^T, \zeta_1^T) = \mathbb{1}\{f(X_1^T) = g(\zeta_1^T)\}$. Similar the proof technique of Theorem 2, it can be shown that $C(k)$ in Appendix K cannot equal $(\sum_{x_1^T : f(x_1^T) = k} P_{X_1^T}(x_1^T)) \wedge \alpha$ for all $k \in [K]$, while the worst-case $f$-robust Type-I error remains upper bounded by $\alpha$ for all $Q_{X_1^T}$ and $\epsilon$.

- **Case 3:** Let $\Xi_\gamma(x_1^T) := \{\zeta_1^T \in \mathcal{Z}^T : \gamma(x_1^T, \zeta_1^T) = 1\}$. $\exists x_1^T, y_1^T \in \mathcal{V}^T$, s.t. $f(x_1^T) \neq f(y_1^T)$ and $\Xi_\gamma(x_1^T) \cap \Xi_\gamma(y_1^T) \neq \emptyset$. For any detector $\gamma \notin \Gamma_f^*$ that does not belong to Cases 1 and 2, it belongs to Case 3. Let us start from a simple case where $T = 1$, $\mathcal{V} = \{x_1, x_2, x_3\}$, $K = 2$, $\mathcal{Z} = \{\zeta_1, \zeta_2, \zeta_3\}$, and $\mathcal{S} = [2]$. Consider the mapping $f$ and the detector as follows: $f(x_1) = f(x_2) = 1$, $f(x_3) = 2$, $\gamma(x_1, \zeta_1) = \gamma(x_1, \zeta_1) = 1$, $\gamma(x_3, \zeta_2) = 1$, and $\gamma(x, \zeta) = 0$ for all other pairs $(x, \zeta)$. When $C(k) = (\sum_{x_1^T : f(x_1^T) = k} P_{X_1^T}(x_1^T)) \wedge \alpha$ for all $k \in [K]$, i.e.,
$$P_{X, \zeta}(x_1, \zeta_1) + P_{X, \zeta}(x_1, \zeta_2) + P_{X, \zeta}(x_2, \zeta_1) + P_{X, \zeta}(x_2, \zeta_2) = (P_X(x_1) + P_X(x_2)) \wedge \alpha,$$
and $\quad P_{X, \zeta}(x_3, \zeta_2) = P_X(x_3) \wedge \alpha,$

then the worst-case $f$-robust Type-I error is lower bounded by

$$\max_{Q_{X_1^T}} \mathbb{E}_{Q_{X_1^T} \otimes P_{\zeta_1^T}} \left[ \sup_{\tilde{x}_1^T \in \mathcal{B}_f(X_1^T)} \mathbb{1}\{\gamma(\tilde{x}_1^T, \zeta_1^T) = 1\} \right]$$

$$\geq \mathbb{E}_{P_{\zeta_1^T}} \left[ \sup_{\tilde{x}_1^T \in \mathcal{B}(1)} \mathbb{1}\{\gamma(\tilde{x}_1^T, \zeta_1^T) = 1\} \right]$$

$$= (P_X(x_1) + P_X(x_2)) \wedge \alpha + P_X(x_3) \wedge \alpha$$

$$> \alpha, \quad \text{if } P_X(x_1) + P_X(x_2) > \alpha \text{ or } P_X(x_3) > \alpha.$$

Thus, for any $Q_X$ such that $\{P_X \in \mathcal{P}(\mathcal{V}) : \mathsf{D}_{\mathsf{TV}}(P_X, Q_X) \leq \epsilon\} \subseteq \{P_X \in \mathcal{P}(\mathcal{V}) : P_X(x_1) + P_X(x_2) > \alpha \text{ or } P_X(x_2) > \alpha\}$, the false-alarm constraint is violated when $C(k) = (\sum_{x_1^T : f(x_1^T) = k} P_{X_1^T}(x_1^T)) \wedge \alpha$ for all $k \in [K]$. The result can be generalized to larger $(T, \mathcal{V}, \mathcal{Z}, K, \mathcal{S})$, other functions $f$, and other detectors that belong to Case 3.

In conclusion, if and only if $\gamma \in \Gamma^*$, the minimum Type-II error attained from (Opt-R) reaches the universal minimum $f$-robust Type-II error $\beta_1^*(f, Q_{X_1^T}, \epsilon, \alpha)$ in (14) for all $Q_{X_1^T} \in \mathcal{P}(\mathcal{V}^T)$ and $\epsilon \in \mathbb{R}_{\geq 0}$.

Under the watermarking scheme $P_{X_1^T, \zeta_1^T}^{*, f}$, the $f$-robust Type-I and Type-II errors are given by:

**$f$-robust Type-I error:**

$$\because \forall y_1^T \in \mathcal{V}^T, \quad \mathbb{E}_{P_{\zeta_1^T}^{*, f}} \left[ \sup_{\tilde{x}_1^T \in \mathcal{B}_f(y_1^T)} \mathbb{1}\{\gamma(\tilde{x}_1^T, \zeta_1^T) = 1\} \right]$$

$$= \sum_{\zeta_1^T} \sum_{x_1^T} P_{X_1^T, \zeta_1^T}^{*, f}(x_1^T, \zeta_1^T) \sup_{\tilde{x}_1^T \in \mathcal{B}_f(y_1^T))} \mathbb{1}\{\gamma(\tilde{x}_1^T, \zeta_1^T) = 1\}$$

$$= \sum_{x_1^T \in \mathcal{B}_f(y_1^T)} P_{X_1^T}^{*, f}(x_1^T) \sum_{\zeta_1^T} P_{\zeta_1^T | X_1^T}^{*, f}(\zeta_1^T | x_1^T) \sup_{\tilde{x}_1^T \in \mathcal{B}_f(y_1^T)} \mathbb{1}\{\gamma(\tilde{x}_1^T, \zeta_1^T) = 1\}$$

$$= \left( \sum_{x_1^T \in \mathcal{B}_f(y_1^T)} P_{X_1^T}^{*, f}(x_1^T) \right) \wedge \alpha \leq \alpha,$$

and since any distribution $Q_{X_1^T}$ can be written as a linear combinations of $\delta_{y_1^T}$,

$$\therefore \sup_{Q_{X_1^T}} \mathbb{E}_{Q_{X_1^T} P_{\zeta_1^T}^{*, f}} \left[ \sup_{\tilde{x}_1^T \in \mathcal{B}_f(X_1^T)} \mathbb{1}\{\gamma(\tilde{x}_1^T, \zeta_1^T) = 1\} \right] \leq \alpha.$$

**$f$-robust Type-II error:**

$$1 - \mathbb{E}_{P_{X_1^T, \zeta_1^T}^{*, f}} \left[ \sup_{\tilde{x}_1^T \in \mathcal{B}_f(X_1^T)} \mathbb{1}\{\gamma(\tilde{x}_1^T, \zeta_1^T) = 1\} \right]$$

$$= 1 - \sum_{x_1^T} \sum_{\zeta_1^T} P_{X_1^T, \zeta_1^T}^{*, f}(x_1^T, \zeta_1^T) \sup_{\tilde{x}_1^T \in \mathcal{B}_f(x_1^T)} \mathbb{1}\{\gamma(\tilde{x}_1^T, \zeta_1^T) = 1\}$$

$$= 1 - \sum_{k \in [K]} \sum_{x_1^T \in \mathcal{B}(k)} P_{X_1^T}^{*, f}(x_1^T) \sum_{\zeta_1^T} P_{\zeta_1^T | X_1^T}^{*, f}(\zeta_1^T | x_1^T) \sup_{\tilde{x}_1^T \in \mathcal{B}(k)} \mathbb{1}\{\gamma(\tilde{x}_1^T, \zeta_1^T) = 1\}$$

$$= 1 - \sum_{k \in [K]} \left( \left( \sum_{x_1^T \in \mathcal{B}(k)} P_{X_1^T}^{*, f}(x_1^T) \right) \wedge \alpha \right)$$

$$= \sum_{k \in [K]} \left( \left( \sum_{x_1^T \in \mathcal{B}(k)} P_{X_1^T}^{*, f}(x_1^T) \right) - \alpha \right)_+.$$

The optimality of $P_{X_1^T, \zeta_1^T}^{*, f}$ is thus proved. $\qquad\square$

Figure 6 compares the universally minimum Type-II errors with and without semantic-invariant text modification.

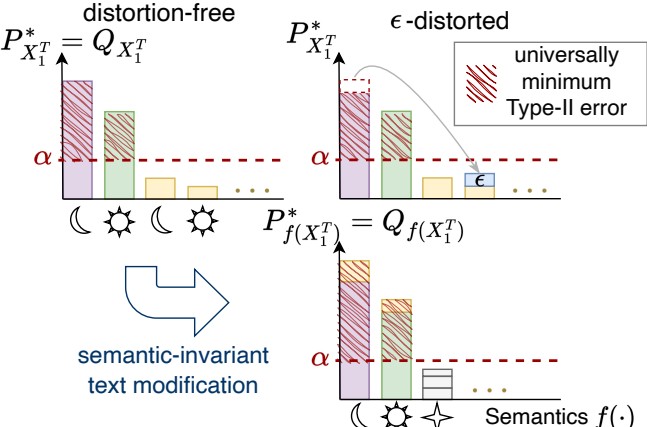

Figure 6: Universally minimum Type-II error w/o distortion and with semantic-invariant text modification.

# M    Broader Impacts

This paper introduces a novel framework and algorithm for LLM watermarking, aimed at advancing the field of machine learning by enhancing AI safety and data authenticity. The primary positive impacts of our work include its potential to identify AI-generated misinformation, ensure integrity in academia and society, protect intellectual property (IP), and enhance public trust in AI technologies. However, given the dual-use potential of watermarking techniques, it is crucial to consider privacy concerns raised by possible misuse–such as unauthorized tracking of data. We encourage the development of ethical guidelines to ensure the responsible use and deployment of this technology. By considering both beneficial outcomes and potential risks, our work seeks to contribute responsibly to the machine learning community.

# NeurIPS Paper Checklist

1. **Claims**

   Question: Do the main claims made in the abstract and introduction accurately reflect the paper's contributions and scope?

   Answer: [Yes]

   Justification: The abstract and introduction accurately summarize the paper's contributions, including the development of a unified theoretical framework for joint watermark scheme-detector optimization, derivation of optimal solutions, the DAWA algorithm, and experimental validation. These claims are consistent with the content presented in the subsequent sections and appendices.

   Guidelines:

   - The answer NA means that the abstract and introduction do not include the claims made in the paper.
   - The abstract and/or introduction should clearly state the claims made, including the contributions made in the paper and important assumptions and limitations. A No or NA answer to this question will not be perceived well by the reviewers.
   - The claims made should match theoretical and experimental results, and reflect how much the results can be expected to generalize to other settings.
   - It is fine to include aspirational goals as motivation as long as it is clear that these goals are not attained by the paper.

2. **Limitations**

   Question: Does the paper discuss the limitations of the work performed by the authors?

   Answer: [Yes]

   Justification: The paper discusses "Practical Challenges" in implementing the theoretically optimal scheme (Section 3, page 5, line 225), which motivates the token-level design. Section 6.2 and Appendix J discuss the robustness of DAWA, including its performance against certain attacks like deletion and paraphrasing (empirically addressed in Appendix I, Table 9) and outlines theoretical extensions for stronger future robustness, implying current trade-offs. A specific acknowledged limitation is that the sensitivity of DAWA to surrogate model mismatch in extreme cases was not empirically tested, although Appendix I shows robustness to varied SLMs in non-extreme scenarios.

   Guidelines:

   - The answer NA means that the paper has no limitation while the answer No means that the paper has limitations, but those are not discussed in the paper.
   - The authors are encouraged to create a separate "Limitations" section in their paper.
   - The paper should point out any strong assumptions and how robust the results are to violations of these assumptions (e.g., independence assumptions, noiseless settings, model well-specification, asymptotic approximations only holding locally). The authors should reflect on how these assumptions might be violated in practice and what the implications would be.
   - The authors should reflect on the scope of the claims made, e.g., if the approach was only tested on a few datasets or with a few runs. In general, empirical results often depend on implicit assumptions, which should be articulated.
   - The authors should reflect on the factors that influence the performance of the approach. For example, a facial recognition algorithm may perform poorly when image resolution is low or images are taken in low lighting. Or a speech-to-text system might not be used reliably to provide closed captions for online lectures because it fails to handle technical jargon.
   - The authors should discuss the computational efficiency of the proposed algorithms and how they scale with dataset size.
   - If applicable, the authors should discuss possible limitations of their approach to address problems of privacy and fairness.

- While the authors might fear that complete honesty about limitations might be used by reviewers as grounds for rejection, a worse outcome might be that reviewers discover limitations that aren't acknowledged in the paper. The authors should use their best judgment and recognize that individual actions in favor of transparency play an important role in developing norms that preserve the integrity of the community. Reviewers will be specifically instructed to not penalize honesty concerning limitations.

3. **Theory assumptions and proofs**

   Question: For each theoretical result, does the paper provide the full set of assumptions and a complete (and correct) proof?

   Answer: [Yes]

   Justification: Key theoretical results (Theorems 1, 2, 6, 7; Lemma 3; Proposition 4) are presented with their assumptions stated. The paper indicates that complete proofs are provided in the corresponding appendices (e.g., Appendix C for Theorem 1, Appendix D for Theorem 2, Appendix K for Theorem 6, Appendix L for 7, Appendix G for Lemma 3), following standard practice.

   Guidelines:

   - The answer NA means that the paper does not include theoretical results.
   - All the theorems, formulas, and proofs in the paper should be numbered and cross-referenced.
   - All assumptions should be clearly stated or referenced in the statement of any theorems.
   - The proofs can either appear in the main paper or the supplemental material, but if they appear in the supplemental material, the authors are encouraged to provide a short proof sketch to provide intuition.
   - Inversely, any informal proof provided in the core of the paper should be complemented by formal proofs provided in appendix or supplemental material.
   - Theorems and Lemmas that the proof relies upon should be properly referenced.

4. **Experimental result reproducibility**

   Question: Does the paper fully disclose all the information needed to reproduce the main experimental results of the paper to the extent that it affects the main claims and/or conclusions of the paper (regardless of whether the code and data are provided or not)?

   Answer: [Yes]

   Justification: Section 6 (Experiments) and Appendix I detail the experimental settings, including models used (Llama2-13B, Mistral-8×7B, surrogate models), datasets (C4, ELI5, Wikipedia), prompts, DAWA parameters $(\eta, T)$, baselines, and evaluation metrics. This should provide sufficient information for others to understand and attempt to reproduce the core findings.

   Guidelines:

   - The answer NA means that the paper does not include experiments.
   - If the paper includes experiments, a No answer to this question will not be perceived well by the reviewers: Making the paper reproducible is important, regardless of whether the code and data are provided or not.
   - If the contribution is a dataset and/or model, the authors should describe the steps taken to make their results reproducible or verifiable.
   - Depending on the contribution, reproducibility can be accomplished in various ways. For example, if the contribution is a novel architecture, describing the architecture fully might suffice, or if the contribution is a specific model and empirical evaluation, it may be necessary to either make it possible for others to replicate the model with the same dataset, or provide access to the model. In general. releasing code and data is often one good way to accomplish this, but reproducibility can also be provided via detailed instructions for how to replicate the results, access to a hosted model (e.g., in the case of a large language model), releasing of a model checkpoint, or other means that are appropriate to the research performed.

- While NeurIPS does not require releasing code, the conference does require all submissions to provide some reasonable avenue for reproducibility, which may depend on the nature of the contribution. For example
    (a) If the contribution is primarily a new algorithm, the paper should make it clear how to reproduce that algorithm.
    (b) If the contribution is primarily a new model architecture, the paper should describe the architecture clearly and fully.
    (c) If the contribution is a new model (e.g., a large language model), then there should either be a way to access this model for reproducing the results or a way to reproduce the model (e.g., with an open-source dataset or instructions for how to construct the dataset).
    (d) We recognize that reproducibility may be tricky in some cases, in which case authors are welcome to describe the particular way they provide for reproducibility. In the case of closed-source models, it may be that access to the model is limited in some way (e.g., to registered users), but it should be possible for other researchers to have some path to reproducing or verifying the results.

5. **Open access to data and code**

   Question: Does the paper provide open access to the data and code, with sufficient instructions to faithfully reproduce the main experimental results, as described in supplemental material?

   Answer: [No]

   Justification: We are committed to open science and will make the code for our DAWA algorithm and experimental scripts publicly available on GitHub upon publication of this paper. The datasets used in our experiments (C4, ELI5, Wikipedia) are already publicly available and are cited in the manuscript. While an anonymized version of our code is not available at the time of submission, the paper provides detailed pseudo-code for DAWA (Appendix H) and a thorough description of our experimental setup (Section 6 and Appendix I) to allow for a detailed understanding and facilitate future reproduction.

   Guidelines:

   - The answer NA means that paper does not include experiments requiring code.
   - Please see the NeurIPS code and data submission guidelines (`https://nips.cc/public/guides/CodeSubmissionPolicy`) for more details.
   - While we encourage the release of code and data, we understand that this might not be possible, so "No" is an acceptable answer. Papers cannot be rejected simply for not including code, unless this is central to the contribution (e.g., for a new open-source benchmark).
   - The instructions should contain the exact command and environment needed to run to reproduce the results. See the NeurIPS code and data submission guidelines (`https://nips.cc/public/guides/CodeSubmissionPolicy`) for more details.
   - The authors should provide instructions on data access and preparation, including how to access the raw data, preprocessed data, intermediate data, and generated data, etc.
   - The authors should provide scripts to reproduce all experimental results for the new proposed method and baselines. If only a subset of experiments are reproducible, they should state which ones are omitted from the script and why.
   - At submission time, to preserve anonymity, the authors should release anonymized versions (if applicable).
   - Providing as much information as possible in supplemental material (appended to the paper) is recommended, but including URLs to data and code is permitted.

6. **Experimental setting/details**

   Question: Does the paper specify all the training and test details (e.g., data splits, hyperparameters, how they were chosen, type of optimizer, etc.) necessary to understand the results?

   Answer: [Yes]

Justification: Section 6 ("Experiment Settings") and Appendix I provide details on models, datasets, prompts, watermark parameters (e.g., $\eta = 0.2, T = 200$ for DAWA, strength for KGW+23), and evaluation metrics. The choice of $\eta$ is linked to theoretical considerations in Lemma 3 and Appendix G.

Guidelines:

- The answer NA means that the paper does not include experiments.
- The experimental setting should be presented in the core of the paper to a level of detail that is necessary to appreciate the results and make sense of them.
- The full details can be provided either with the code, in appendix, or as supplemental material.

7. **Experiment statistical significance**

Question: Does the paper report error bars suitably and correctly defined or other appropriate information about the statistical significance of the experiments?

Answer: [No]

Justification: The paper reports primary evaluation metrics such as ROC-AUC and True Positive Rates (TPR) at specific False Positive Rates (FPRs), with detailed explanations of these metrics provided in Section 6. While error bars, confidence intervals, or formal statistical significance tests are not included for these reported values, our experiments are conducted on large, fixed test sets (e.g., $10^5$ texts for ultra-low FPR analysis, as described in Section 6), which provides stability to the point estimates of these metrics. The observed performance differences, as shown in Figure 1 and Table 1, are substantial. We acknowledge that quantifying variability, for instance, through multiple runs or bootstrapping, could further substantiate the comparisons, though it is often not standard practice for ROC-AUC and TPR reporting in this specific evaluation context due to the nature of these aggregate metrics on large test corpora and the computational demands of re-running experiments with large language models.

Guidelines:

- The answer NA means that the paper does not include experiments.
- The authors should answer "Yes" if the results are accompanied by error bars, confidence intervals, or statistical significance tests, at least for the experiments that support the main claims of the paper.
- The factors of variability that the error bars are capturing should be clearly stated (for example, train/test split, initialization, random drawing of some parameter, or overall run with given experimental conditions).
- The method for calculating the error bars should be explained (closed form formula, call to a library function, bootstrap, etc.)
- The assumptions made should be given (e.g., Normally distributed errors).
- It should be clear whether the error bar is the standard deviation or the standard error of the mean.
- It is OK to report 1-sigma error bars, but one should state it. The authors should preferably report a 2-sigma error bar than state that they have a 96% CI, if the hypothesis of Normality of errors is not verified.
- For asymmetric distributions, the authors should be careful not to show in tables or figures symmetric error bars that would yield results that are out of range (e.g. negative error rates).
- If error bars are reported in tables or plots, The authors should explain in the text how they were calculated and reference the corresponding figures or tables in the text.

8. **Experiments compute resources**

Question: For each experiment, does the paper provide sufficient information on the computer resources (type of compute workers, memory, time of execution) needed to reproduce the experiments?

Answer: [Yes]

Justification: Section 6 mentions the use of "Nvidia A100 GPUs". Appendix I (Table 4) provides average generation time comparisons for watermarked and unwatermarked text, giving an indication of the computational overhead of the watermarking process. This provides a baseline for understanding compute requirements.

Guidelines:

- The answer NA means that the paper does not include experiments.
- The paper should indicate the type of compute workers CPU or GPU, internal cluster, or cloud provider, including relevant memory and storage.
- The paper should provide the amount of compute required for each of the individual experimental runs as well as estimate the total compute.
- The paper should disclose whether the full research project required more compute than the experiments reported in the paper (e.g., preliminary or failed experiments that didn't make it into the paper).

9. **Code of ethics**

Question: Does the research conducted in the paper conform, in every respect, with the NeurIPS Code of Ethics https://neurips.cc/public/EthicsGuidelines?

Answer: [Yes]

Justification: The research aims to develop methods for identifying AI-generated text, which can help mitigate misuse such as disinformation and plagiarism, aligning with ethical goals of responsible AI. We have strived for integrity in our methodology, theoretical derivations, experimental procedures, and reporting of results. The datasets used are publicly available and appropriately cited, and the research did not involve direct human experimentation that would require IRB approval beyond standard dataset usage. However, given the dual-use potential of watermarking techniques, it is crucial to consider privacy concerns raised by possible misuse–such as unauthorized tracking of data. We encourage the development of ethical guidelines to ensure responsible use and deployment of this technology.

Guidelines:

- The answer NA means that the authors have not reviewed the NeurIPS Code of Ethics.
- If the authors answer No, they should explain the special circumstances that require a deviation from the Code of Ethics.
- The authors should make sure to preserve anonymity (e.g., if there is a special consideration due to laws or regulations in their jurisdiction).

10. **Broader impacts**

Question: Does the paper discuss both potential positive societal impacts and negative societal impacts of the work performed?

Answer: [Yes]

Justification: The paper discusses societal impacts. Section 1 outlines the significant positive societal impacts, such as mitigating AI-driven disinformation and enhancing data authenticity. The paper also acknowledges challenges related to watermark robustness and the evolving nature of attacks in Section 6.2 and Appendix J. These inherent challenges in ensuring perfect, unbreakable watermarks implicitly relate to potential negative outcomes if the technology is misused or circumvented. The discussion in Appendix M also addresses the dual-use nature of watermarking, acknowledging potential negative impacts such as privacy concerns from misuse (e.g., unauthorized data tracking). In light of these risks, the paper underscores the importance of developing ethical guidelines for responsible deployment and use of this technology.

Guidelines:

- The answer NA means that there is no societal impact of the work performed.
- If the authors answer NA or No, they should explain why their work has no societal impact or why the paper does not address societal impact.
- Examples of negative societal impacts include potential malicious or unintended uses (e.g., disinformation, generating fake profiles, surveillance), fairness considerations (e.g., deployment of technologies that could make decisions that unfairly impact specific groups), privacy considerations, and security considerations.

- The conference expects that many papers will be foundational research and not tied to particular applications, let alone deployments. However, if there is a direct path to any negative applications, the authors should point it out. For example, it is legitimate to point out that an improvement in the quality of generative models could be used to generate deepfakes for disinformation. On the other hand, it is not needed to point out that a generic algorithm for optimizing neural networks could enable people to train models that generate Deepfakes faster.
- The authors should consider possible harms that could arise when the technology is being used as intended and functioning correctly, harms that could arise when the technology is being used as intended but gives incorrect results, and harms following from (intentional or unintentional) misuse of the technology.
- If there are negative societal impacts, the authors could also discuss possible mitigation strategies (e.g., gated release of models, providing defenses in addition to attacks, mechanisms for monitoring misuse, mechanisms to monitor how a system learns from feedback over time, improving the efficiency and accessibility of ML).

11. **Safeguards**

Question: Does the paper describe safeguards that have been put in place for responsible release of data or models that have a high risk for misuse (e.g., pretrained language models, image generators, or scraped datasets)?

Answer: [NA]

Justification: The paper introduces a watermarking methodology and an algorithm (DAWA). It does not release new large language models or novel datasets scraped from the internet that would pose a high risk for misuse. The models used for experiments (Llama2, Mistral) are existing models developed by other parties.

Guidelines:

- The answer NA means that the paper poses no such risks.
- Released models that have a high risk for misuse or dual-use should be released with necessary safeguards to allow for controlled use of the model, for example by requiring that users adhere to usage guidelines or restrictions to access the model or implementing safety filters.
- Datasets that have been scraped from the Internet could pose safety risks. The authors should describe how they avoided releasing unsafe images.
- We recognize that providing effective safeguards is challenging, and many papers do not require this, but we encourage authors to take this into account and make a best faith effort.

12. **Licenses for existing assets**

Question: Are the creators or original owners of assets (e.g., code, data, models), used in the paper, properly credited and are the license and terms of use explicitly mentioned and properly respected?

Answer: [Yes]

Justification: The paper properly cites the original sources for datasets (C4 [49], ELI5 [50], WMT19 dataset [52]) and models (Llama2-13B [1], Mistral-8×7B [44], GPT-3 [51], MBART [53]). The licenses and terms of use for these assets are explicitly mentioned (Appendix I) and properly respected.

Guidelines:

- The answer NA means that the paper does not use existing assets.
- The authors should cite the original paper that produced the code package or dataset.
- The authors should state which version of the asset is used and, if possible, include a URL.
- The name of the license (e.g., CC-BY 4.0) should be included for each asset.
- For scraped data from a particular source (e.g., website), the copyright and terms of service of that source should be provided.

- If assets are released, the license, copyright information, and terms of use in the package should be provided. For popular datasets, `paperswithcode.com/datasets` has curated licenses for some datasets. Their licensing guide can help determine the license of a dataset.
- For existing datasets that are re-packaged, both the original license and the license of the derived asset (if it has changed) should be provided.
- If this information is not available online, the authors are encouraged to reach out to the asset's creators.

13. **New assets**

Question: Are new assets introduced in the paper well documented and is the documentation provided alongside the assets?

Answer: [NA]

Justification: The paper introduces a new theoretical framework and algorithm (DAWA), which are documented within the paper itself (Appendix H for pseudo-code). No separate new datasets or software packages are being released that would require external documentation beyond the paper.

Guidelines:

- The answer NA means that the paper does not release new assets.
- Researchers should communicate the details of the dataset/code/model as part of their submissions via structured templates. This includes details about training, license, limitations, etc.
- The paper should discuss whether and how consent was obtained from people whose asset is used.
- At submission time, remember to anonymize your assets (if applicable). You can either create an anonymized URL or include an anonymized zip file.

14. **Crowdsourcing and research with human subjects**

Question: For crowdsourcing experiments and research with human subjects, does the paper include the full text of instructions given to participants and screenshots, if applicable, as well as details about compensation (if any)?

Answer: [NA]

Justification: The research does not involve crowdsourcing or direct human subject experiments (e.g., user studies with participant recruitment). Evaluation of text quality uses automated metrics (perplexity, BLEU) and existing datasets of human/AI text.

Guidelines:

- The answer NA means that the paper does not involve crowdsourcing nor research with human subjects.
- Including this information in the supplemental material is fine, but if the main contribution of the paper involves human subjects, then as much detail as possible should be included in the main paper.
- According to the NeurIPS Code of Ethics, workers involved in data collection, curation, or other labor should be paid at least the minimum wage in the country of the data collector.

15. **Institutional review board (IRB) approvals or equivalent for research with human subjects**

Question: Does the paper describe potential risks incurred by study participants, whether such risks were disclosed to the subjects, and whether Institutional Review Board (IRB) approvals (or an equivalent approval/review based on the requirements of your country or institution) were obtained?

Answer: [NA]

Justification: As no direct human subject research or crowdsourcing was conducted (see Q14), IRB approval was not applicable.

Guidelines:

- The answer NA means that the paper does not involve crowdsourcing nor research with human subjects.
- Depending on the country in which research is conducted, IRB approval (or equivalent) may be required for any human subjects research. If you obtained IRB approval, you should clearly state this in the paper.
- We recognize that the procedures for this may vary significantly between institutions and locations, and we expect authors to adhere to the NeurIPS Code of Ethics and the guidelines for their institution.
- For initial submissions, do not include any information that would break anonymity (if applicable), such as the institution conducting the review.

16. **Declaration of LLM usage**

Question: Does the paper describe the usage of LLMs if it is an important, original, or non-standard component of the core methods in this research? Note that if the LLM is used only for writing, editing, or formatting purposes and does not impact the core methodology, scientific rigorousness, or originality of the research, declaration is not required.

Answer: [Yes]

Justification: The proposed DAWA algorithm, specifically its detection phase, utilizes a Surrogate Language Model (SLM) as a component to approximate the watermarked distributions and enable model-agnostic detection (Section 5 "Novel Tricks"); Algorithm 2). This use of an LLM (the SLM) is integral to the methodology of the DAWA detector.

Guidelines:

- The answer NA means that the core method development in this research does not involve LLMs as any important, original, or non-standard components.
- Please refer to our LLM policy (`https://neurips.cc/Conferences/2025/LLM`) for what should or should not be described.

