# OpenReview forum: "Theoretically Grounded Framework for LLM Watermarking: A Distribution-Adaptive Approach"
_NeurIPS.cc/2025/Conference — NeurIPS 2025 poster_

### Official Review · Reviewer_eRkN · 2025-06-27

**Clarity:** 3
**Significance:** 3
**Originality:** 3
**Rating:** 5
**Confidence:** 4

**Summary:**

This paper studied the problem of watermarking and its detection. Authors first formulate this problem in a hypothesis testing framework, aiming to jointly find the optimal watermarking scheme and the corresponding detector. In particular, an optimal detector achieves the highest power under a type I error constraint, while the watermarking scheme satisfies a distribution distrotion constraint. Authors provide a lower bound for the detector's type II error and an detector that achieve this bound. In addition, a practical algorithm DAWA is proposed and demonstrated better performance (large AUC-ROC score) compared to baseline methods.

**Questions:**

Please see the Concerns section above.

**Ethical Concerns:**

["NO or VERY MINOR ethics concerns only"]

**Final Justification:**

Authors have addressed my concern on the reconstruction process, and explained the benefit of using DAWA compared to other token-level watermarking methods. I am therefore leaning to acceptance.

**Limitations:**

Yes

**Quality:**

3

**Strengths And Weaknesses:**

**Strengths:**

1. The motivation and main messages are clearly explained and easy to follow. By formulating the problem in the classic hypothesis testing framework, Section 3 seems to provide a clever and complete characterization for performance of watermarking detectors.

2. The topic of watermarking and its detection is definitely timely and important.

3. The insights sound right to me, such that a good scheme should be adaptive to LLMs.



**Concerns:**

I have no major concerns about this work, below are some minor comments:

1. The algorithm clearly has a gap compared to the theoretical results, since some key assumptions do not hold in practice. Authors acknowledged those challenges and proposed DAWA as an approximation. In particular, my questions are:

   (i) DAWA can be sub-optimal in the sentence level. Lemma 3 and Proposition 4 are not unique properies for DAWA, in fact, other token-level detectors such as those in Example 2 also enjoy these nice properties (due to concentration brought by IID $\zeta_t$). Therefore, it is unclear how much DAWA is better than existing designs from a theoretical perspective. While authors have empricially shown the superiority of DAWA, I feel it still has to been clarified.

  (ii) The recovery process of $\zeta_t$. Why a significantly different SLM than the original model can accurately reconstruct $\zeta_t$? My impression is that the recovery can be successful (even partially) when the surrogate model is similar to the original LLM.

---

> ### Author Rebuttal · Authors · 2025-07-31
>
> We are grateful for your time, valuable feedback, and appreciation of our theoretical contributions. In the following, we address your concerns point by point.
>
> >**Q1.** DAWA can be sub-optimal at the sentence level. Lemma 3 and Proposition 4 are not unique properties for DAWA, in fact, other token-level detectors, such as those in Example 2 also enjoy these nice properties (due to concentration brought by IID \\zeta\_t). Therefore, it is unclear how much DAWA is better than existing designs from a theoretical perspective. While authors have empirically shown the superiority of DAWA, I feel it still has to be clarified.
>
> **A1.** Thanks for the insightful comment. It is correct that DAWA is a **token-level adaptation** of the **jointly optimal sentence-level** watermarking and detection scheme. In **Lemma 3** and **Proposition 4**, our goal is to show that **key theoretical guarantees of the sentence-level optimal design can be partially preserved** in the token-level setting through DAWA, **a property not shared by other token-level methods** such as those in Examples 1 and 2\.
>
> Specifically:
>
> * **Lemma 3** demonstrates that **DAWA uniquely controls the worst-case sentence-level FPR** by regulating the token-level FPR. This is **non-trivial** and relies on DAWA’s design being **adaptive to the LLM’s next-token prediction (NTP) distribution**. Such control is **not guaranteed** for other token-level detectors.
> * **Proposition 4** establishes **robustness to token replacements**, which results from DAWA’s **design of redundant auxiliary variable $\\tilde{\\zeta}$**. This auxiliary redundancy is a **distinctive design element** of DAWA, not present in prior schemes.
>
> While we agree that these results are not directly comparable to theoretical guarantees of other schemes under different formulations, our theoretical analysis complements **extensive experiments** that validate Lemma 3 and Proposition 4 and directly compare DAWA to strong baselines.
>
> In summary, DAWA is both **principled and practical**: it inherits desirable properties from an optimal sentence-level formulation while remaining effective in the real-world token-level context. We believe this combination, supported by theory and validated by experiments, justifies its contribution. We will clarify this point further in the revised version and continue to refine DAWA both theoretically and empirically.
>
> ---
> >**Q2.** The recovery process of $\zeta\_t$. Why a significantly different SLM than the original model can accurately reconstruct $\zeta\_t$? My impression is that the recovery can be successful (even partially) when the surrogate model is similar to the original LLM.
>
> **A2.** Thank you for the sharp observation. While it may seem intuitive that accurately recovering $\\zeta\_t$ would require an SLM similar to the original LLM, our results demonstrate otherwise; **DAWA’s recovery process of auxiliary variables is robust even with a different SLM**.
>
> As shown in **Appendix K, Tables 4 and 5**, accurate recovery is still achievable using a **much smaller SLM from a different model family**, and even **without prompts**. This robustness stems from the **special structure of the conditional distribution** $P\_{\\zeta\_t \\mid x\_1^t}$ induced by the **optimal watermarking design**.
>
> In particular:
>
> * As illustrated in **Figure 4**, recovering $\\zeta\_t$ only requires reconstructing the conditional distribution $P\_{\\zeta\_t \\mid x\_1^t}$ and does not require the surrogate NTP distribution $\\tilde{Q}\_{X\_t \\mid x\_1^{t-1}}$ to precisely match the true token distribution.
> * As shown in the **right-hand side of Figure 5 (Appendix F)**, for **auxiliary values corresponding to high-probability tokens**, their probability under $P\_{\\zeta\_t \\mid x\_1^t}$ is **set to the fixed FPR control parameter** $\\eta$, and is therefore **insensitive to SLM mismatches**.
> * For **low-probability tokens**, the corresponding auxiliary value probability **directly reflects the token’s likelihood**, which is typically **low and less sensitive** to variation across models.
> * Furthermore, the **redundant auxiliary variable** $\\tilde{\\zeta}$ is assigned a **significantly higher probability** than the other values, making it robust to small perturbations in $Q\_{X\_t \\mid x\_1^{t-1}}$ and improving the overall recovery accuracy.
>
> Thus, DAWA’s recovery of auxiliary variables is **not critically dependent** on having a close surrogate model. This robustness is a **key strength** of our design, and we will clarify this point further in the final version.
>
> ---
> We sincerely appreciate your support and thoughtful review. We hope our responses have adequately addressed your concerns.

---

> > ### Comment · Reviewer_eRkN · 2025-08-02
> >
> > I thank the authors for their responses, particularly the insights on watermark dectection. However, my first concern is not fully addressed. As I originally mentioned, Lemma 3 and Prop 4 are not unique properties of DAWA. Other token-level detectors, such as those studied in Li et al., 2025, can obtain similar results simply by applying concentration inequalities. I agreed that DAWA seems to outperform other methods empirically, though it is unclear whether DAWA achieves a better performance from a theoretical perspective.
> >
> >
> > References:
> >
> > Li X, Ruan F, Wang H, et al. A statistical framework of watermarks for large language models: Pivot, detection efficiency and optimal rules[J]. The Annals of Statistics, 2025, 53(1): 322-351.

---

> > > ### Author Response · Authors · 2025-08-06
> > >
> > > Thank you for the thoughtful follow-up. You are right that for token-level detectors with test statistics in the form of sample averages, concentration inequalities can yield similar bounds. However, the error exponents in our result and in Li et al. (2025) are derived under different settings.
> > >
> > > Our setting is more **stringent**: we aim to control the **worst-case** sequence-level FPR under $\\alpha$ across all unwatermarked (i.e., human-written) texts, rather than analyzing performance under specific assumptions on text distributions. This worst-case guarantee is practically important for applications requiring high reliability.
> > >
> > > We also note that the framework in Li et al. (2025) solves a **surrogate hypothesis testing problem** based on pivotal statistics, **assuming a specific class of text distributions**. They claim that in their problem setting, any detection rule would become powerless if nothing is known about the text distributions. However, this restrictive claim does not hold in our paper. Our design **makes no assumptions about the text distribution** and ensures FPR guarantees over *all* sequences. These differences make a direct theoretical comparison of Type-II errors challenging without special assumptions.
> > >
> > > Moreover, unlike the i.i.d. pivotal statistics in Li et al. (2025) that generally lead to tighter concentration bounds, our setting involves **non-i.i.d. token-level statistics** due to adaptivity to the generative distribution.
> > >
> > > Given these facts above, our concentration bound may be looser than those in the conventional setup with the same $\alpha$. However, **our detector and watermarking scheme remain optimal at the token level**, potentially compensating for the weaker concentration and resulting in the strong empirical performance observed for DAWA.
> > >
> > > In future work, we plan to quantify the gap between the practical bound in Lemma 3 and the universal Type-II error lower bound in Theorem 1, and to design improved token-level detectors with provable advantages.

---

### Official Review · Reviewer_uUH1 · 2025-07-02

**Clarity:** 3
**Significance:** 3
**Originality:** 3
**Rating:** 5
**Confidence:** 3

**Summary:**

In this paper, the authors propose a complete theoretical characterization of an optimal watermarking scheme. Since the theoretical scheme cannot be instantiated in practice, they also propose a practical scheme approximating the design of the theoretical one.

On the theoretical side, a constrained optimization framework is proposed. The authors try to find the embedder/detector pair that maximizes the power of the detector constrained by a fixed probability of false alarm and distortion on the text probabilities. A universal construction for the embedder and the detector is provided by solving the optimization problem.

However, the framework assumes 1) a non-iterative text generation (text of fixed length)  and 2) direct access to the text distribution of the LLM at both the embedder and detector side. This is impractical, hence the authors develop a method inspired from more classic methods, but guaranteed to be optimal **at the token-level**. The problem of accessing the text distribution is addressed by 1) using the token distribution at each step instead and 2) using a LLM at the detection side to retrieve it. A theoretical robustness bound is provided to justify the use of a smaller/worse LLM than during embedding.

Finally, the practical method is evaluated against over state-of-the-art schemes, showing competitive performance in the tested setting.

To note, the appendix contains a lot of theoretical extension of the main paper results towards more interesting semantic attacks.

**Questions:**

1. In the theoretical part, does the key depend on the generation ? As far as I understand, the random sequence is sampled independently but what about the $g$ function? If that is the case, this theoretically imposes a new key for each text which renders the scheme quite useless in practice?
2. Could the authors improve the experimental setting by: using low FPR ($<1e^{-5}$), sound p-values when possible (see [1]), and report the entropy of each model for each tasks ?
3. From the **Flawed robustness evaluation** weakness, could the authors find the reason for the lack of impact of the attack and fix the problem to get more significant results?
4. *(Minor)* It is unclear how the authors control $\alpha$ in the experimental setting. How is $\lambda$ chosen ? A  systematic way to set $\eta$  and $\lambda$ is provided in the appendix, but I would put it in the main paper as it is the most important parameter of the algorithm. Furthermore, the study of the theoretical vs empirical match of the $P_{FA}$ in the appendix is limited, I would like to see an actual curve over many points, going very low in terms of $P_{FA}$ (at least until $1e{-6}$) since very low $P_{FA}$ is the regime of interest for watermarking


**Score increase or decrease** : Given the current state of the paper, I am leaning towards an accept for the theoretical part. However, I might re-evaluate my score depending on the results of a new, better experimental evaluation of the method, given that the current one does not really allow me to a see a significant gain between schemes. I might increase or decrease my score in this situation depending on these results.

**Ethical Concerns:**

["NO or VERY MINOR ethics concerns only"]

**Final Justification:**

The authors answered my questions regarding the practical implementation of the algorithm. The limitations of the methods were clearly presented and I believe that the theoretical contribution outweighs the limited applicability in practice.
The authors also made effort to provide low-FPR results which I find commendable in the current context of watermarking evaluation.
For these reasons, I am ready to raise my score to an accept for this paper.

**Limitations:**

Nothing to report

**Quality:**

2

**Strengths And Weaknesses:**

## Qualities

**Strong theoretical analysis**: The theoretical framework looks solid and with assumptions and limitations well described (with one caveat -- see the questions). Even though the relevance of the framework is quite limited in practice, I believe the method (joint detector/embedder design) and the results stand by themselves as potentially important for more practical designs. I also note even further interesting derivations the appendix for specific kinds of attacks which is appreciated.


**Overall clear presentation the main paper**: Regarding the presentation of the results, the paper is quite well organized (from theory to practice) -- with useful diagrams to explain the more abstract design steps -- notably the construction of the optimal token distribution which is quite trivial when seen through Figure 3 and 5.

## Weakness

**The punchline falls short**: The main weakness of the paper is its practical application. With all the heavy mathematical work done in the first part of the paper, one could expect a straightforward implementation into an actual watermarking scheme. Sadly the assumptions in the theoretical framework are too stringent: optimality computed on text distributions (instead of tokens), assumptions of a static-fixed length text, abstract design of the mapping between the tokens and the secret. This leads the authors to have to make many compromise with respect to the original goal, going back to a very classic design and optimality at the token level for a specific choice of detector.

**Experimental design could be improved**: The experimental design is somewhat weak. The authors report AUC and TPR at quite high FPR in the main paper. This leads to difference in detectability between algorithms which are insignificant -- the claim that DAWA outperforms most other schemes is thus hard to verify here. Strangely, they provide far better metrics in the Appendix with TRP@$1e{-5}$ but only for a single experiment in Table 6. KGW and Aaronson's scheme seem to use the asymptotic detector (z-score) instead of the exact $p$-values described in [1], which would allow a perfect control of the FPR. Finally, the authors claim to test for both low-entropy and high-entropy tasks. However the entropy of generated text is highly dependent on the model used as shown for example in the appendix J of [2].  The distribution of the entropy at each generation not reported, which does not allow to verify their claim regarding entropy.

 **Flawed robustness evaluation**: The robustness test is limited only to a single replacement attack (using T5) with half of token masked. Seeing the lack of impact on the detectability on any of the tested watermarking schemes, either 1) the FPR is far too high or 2) despite half the tokens being masked, a lot less are actually modified or 3) the authors use the *Self-Hash* hashing function to retrieve the secret during detection -- despite its robustness this hashing function is known to be highly insecure [3] and thus should not be used.

[1] Fernandez, P., Chaffin, A., Tit, K., Chappelier, V., & Furon, T. (2023, December). Three bricks to consolidate watermarks for large language models. In 2023 IEEE International Workshop on Information Forensics and Security (WIFS) (pp. 1-6). IEEE.

[2] Giboulot, E., & Furon, T. (2024, December). WaterMax: breaking the LLM watermark detectability-robustness-quality trade-off. In NeurIPS 2024-38th Conference on Neural Information Processing Systems (pp. 1-34).

[3] Nikola Jovanović, Robin Staab, and Martin Vechev. Watermark stealing in large language models. ICML, 2024

---

> ### Author Rebuttal · Authors · 2025-07-31
>
> We are grateful for your time, valuable feedback, and appreciation of our theoretical contributions. In the following, we address your concerns point by point.
>
> >Q1. The punchline falls short.
>
> **A1.** While our theoretical framework makes simplifying assumptions, such as a fixed text length, these enable **tractable and rigorous characterization** of the jointly optimal watermarking scheme and detector. Despite the idealized setting, it provides a **solid foundation** for more realistic models.
>
> More importantly, the theory motivates a **practical watermarking algorithm, DAWA**, based on the key insight that optimal schemes should be **adaptive to the LLM’s generative distribution**—a feature not explored in prior work and central to DAWA’s strong empirical performance.
>
> As shown in **Figure 1**, DAWA clearly outperforms prior distorted and distortion-free baselines, especially in the **ultra-low FPR regime (e.g., $1e-5$)**, demonstrating that our theoretical insights translate into **practical gains**.
>
> ---
> >Q2. Experimental design could be improved.
>
> **A2.** We thank the reviewer for these suggestions and provide detailed clarifications below:
>
> 1. **On evaluation at ultra-low FPR:**
>    We would like to highlight that **Figure 1 on Page 2** already reports **TPR at ultra-low FPRs (e.g., $1e-5$)** as a key experimental result. In this setting, DAWA consistently **outperforms strong baselines**, including KGW+23 [25], EXP-edit [37], Gumbel-Max [38], and HCW+23 [19], with significantly higher TPR. This evaluation was conducted on $10^5$ **texts (200-length) from Wikipedia dataset**, with further details in Section 6\. **Table 6 in Appendix K** complements this by validating that DAWA’s TPR increases with token length T, confirming the benefit of longer contexts.
> 2. **On experiments with FPR $<1e-5$:**
>    In many existing works, such as [3-7] and the baselines we compare against (KGW+23 [25], EXP-edit [37], Gumbel-Max [38], and HCW+23 [19]), detectability evaluations are typically reported **only down to FPR $=1e-4$ or $1e-3$**. In contrast, our performance evaluation at FPR$=1e-5$ already places our analysis in a **more stringent and practically relevant low-FPR regime**.
>    We agree that evaluating under even lower FPR regimes would strengthen the results. However, such experiments require **substantially larger-scale computation and runtime**, which is unfortunately infeasible within the short rebuttal period. We plan to include these extended evaluations in the final version.
> \
>    [3] Jovanović et al. Watermark Stealing in Large Language Models. ICML, 2024. \
>    [4] Kirchenbauer et al. On the Reliability of Watermarks for Large Language Models. ICLR, 2024\.
>    [5] Hans et al. Spotting LLMs with Binoculars: Zero-Shot Detection of Machine-Generated Text. ICML, 2024\.
>    [6] Wouters, B. Optimizing Watermarks for Large Language Models. ICML, 2024\.
>    [7] Chen et al. WatME: Towards Lossless Watermarking Through Lexical Redundancy. ACL, 2024\.
>
> 3. **On p-value in baselines:**
>    We carefully reviewed [1] and acknowledge that it introduces a **non-asymptotic p-value expression** to better align theoretical and empirical FPR. However, the **detector statistics $S_T$** used in KGW+23 [25] and Gumbel-Max [38] remain unchanged, and the refined p-value merely provides a principled way to **set the threshold**.
>    In our experiments, we evaluate detection performance using the ROC-AUC score, which captures performance across the full range of thresholds and **does not rely on setting a specific detection threshold or computing p-values**. Therefore, comparing AUC scores and TPR at fixed FPR provides a fair and empirical measure of relative performance across methods.
>    Moreover, as shown in **Lemma 3 and Table 7 (Appendix K)**, our theoretical choice of FPR in DAWA tightly upper bounds the empirical FPR, providing an **even stronger theoretical guarantee** for threshold selection than [1], which does not ensure an upper bound on empirical FPR.
> 4. **On entropy reporting:**
>    We estimated the entropy values of the C4 and ELI5 datasets using Llama-2-13B and Mixtral-8$\times$7B, respectively, with the temperature set to 1 for both models. As shown in the table below, entropy varies by model, which is consistent with your observation, and we confirm that the C4 dataset consistently has higher entropy than ELI5 dataset.
>
> |     Model    |   C4  |  ELI5  |
> |:------------:|:-----:|:------:|
> |  Llama-2-13B | 0.547 |  0.272 |
> | Mixtral-8x7B | 1.475 | 1.4268 |
>
>
> ---
> >Q3. Flawed robustness evaluation.
>
>
> **A3.** We address each point as follows:
>
> 1. **On the strength of the T5-based attack:**
>    We note that with 50% of tokens masked, approximately 35% were actually replaced on average by the T5 model, constituting a reasonably strong token replacement attack. This level of perturbation is commonly used in prior works (e.g., KGW+23 [25] and Zhao et al.[26]). If an attack completely rewrites more than half of a sentence, it may begin to significantly alter semantics or text quality, making it less practical. Moreover, as shown in our results, this attack **does degrade detectability for HCW+23**, indicating that the attack is impactful. DAWA’s stable detectability under the same setting reflects the robustness of its design, which includes redundant auxiliary variables.
> 2. **On the limited attack types:**
>    In Section 6.2 and **Appendix L.1 (Table 8)**, we also evaluated DAWA under **random deletion** and **paraphrasing attacks**. DAWA outperforms Gumbel-Max and KGW+23 in deletion robustness and matches their performance under paraphrasing. These results confirm that DAWA remains competitive while balancing robustness, efficiency, and detection accuracy, with the potential to demonstrate a graceful trade-off based on our theoretical analyses.
>
> 3. **On the report of ROC-AUC and FPR:**
>    Due to time constraints, we were unable to add robustness experiments on a larger dataset to explore FPR $<1e-5$. However, we report **ROC-AUC** scores in all experiments to reflect detection performance across the full FPR spectrum, ensuring a fair comparison. We plan to add more experiments in the final version.
> 4. **On theoretical generalization to robustness:**
>    Although robustness is not the core focus of this paper, Appendix L outlines how our theoretical framework and optimal solutions extend to scenarios involving a wide range of attacks, including semantic-invariant attacks. These findings offer valuable insights for designing advanced semantic-based watermarking algorithms that are resilient to such attacks in the future.
> 5. **On hashing and use of Self-Hash:**
>    We clarify that we actually employed the **Left-Hash** function in our experiments. Since cryptographic security is outside the scope of this paper, we did not include an in-depth discussion of hash function security. However, we agree that this is an important consideration and will incorporate a more thorough discussion of secure hashing in future work.
>
> ---
> >Q4. In the theoretical part, does the key depend on the generation? As far as I understand, the random sequence is sampled independently, but what about the g function? If that is the case, this theoretically imposes a new key for each text, which renders the scheme quite useless in practice?
>
> **A4.** As we understand it, this question asks if the **secret key** shared between LLM watermark generation and detection depends on the generation scheme. We clarify that the secret key is fixed and used together with prior tokens to generate a sampling seed. Thus, it does not depend on generation and can be directly shared with the detector.
>
> The function $g$ in the detector is simply a **predefined bijection** between the auxiliary variable set $\mathcal{Z}$ and a superset of the token vocabulary $\mathcal{V}$. It is also **fixed and does not change** across generations.
>
> To offer a concrete toy example of $g$: suppose the token vocabulary is $\mathcal{V}=\lbrace a,b,c \rbrace$. We can define an extended set $\mathcal{S}=\lbrace a,b,c,\\# \rbrace \supset \mathcal{V}$ and a bijective $g\_{tk}$ that maps auxiliary symbols $\mathcal{Z}=\lbrace 1,2,3,4 \rbrace$ to elements in $\mathcal{S}$. This mapping remains static and available to both encoder and detector, and does **not impose a new key per text**.
>
> ---
> >Q5. Minor questions.
>
> **A5.** We address each point as follows.
>
> 1. **On controlling $\alpha$ in experiments:**
>    As stated in Lemma 3 of the main text, the sequence-level FPR $\alpha$ is controlled through the chosen token-level FPR $\eta$, the generated sequence length $T$, and the detection threshold $\lambda$. Specifically, if $\eta\in(0, \min\{ 1, (\alpha/\binom{T}{\lceil T\lambda \rceil})^{\frac{1}{\lceil T\lambda \rceil}}\}]$, the worst-case false alarm for a length-$T$ sequence is upper bounded by $\alpha$.
>
>    In general, given $T$ and $\lambda$, a smaller $\eta$ results in a smaller $\alpha$; given $T$ and $\eta$, a higher $\lambda$ results in a smaller $\alpha$.
>    In practice, we use $\eta \= 0.2$ and $T \= 200$, and select $\lambda$ using **roc\_auc\_score function**, which varies $\lambda$ over a full range to compute empirical FPR and corresponding TPR (see Figure 1, Tables 1, 4, 5, and 6).
>
> 2. **Theoretical vs. empirical FPR and very low FPR regimes:**
>    **Appendix K, Table 7** shows that empirical FPR aligns tightly with the theoretical upper bound down to $1e-5$, confirming **Lemma 3**. While additional tests at even lower FPRs (e.g., $1e-6$) would be informative, the **theoretical guarantee already holds for any target $\alpha$**, and Table 7 serves to empirically validate this.
>    We agree ultra-low FPR regimes are important for high-stakes use cases and will include further evaluations in the final version due to the time limit. We also plan to move some explanations from the appendix into the main paper for better clarity.

---

> > ### Comment · Reviewer_uUH1 · 2025-08-05
> >
> > I thank the reviewer for taking the time to answer my questions in such a detailed way. Notably I thank their patience regarding some of my misunderstandings when reading the paper, it is appreciated. Given the answers and clarifications, I am satisfied and will most likely increase my score to an accept.
> >
> > The following is simply provided for further discussion.
> >
> > > A2. We thank the reviewer for these suggestions and provide detailed clarifications below:
> >
> > I apologize, indeed Figure 1 already provides what I was looking for the low FPR regime. I still believe Table 1 enforces bad practice in the community. I know that authors provide it because it has become "a standard" in the ML watermarking community to work using metric which are not relevant for watermarking (AUC TPR@ high FPR), and not doing so would potentially make them vulnerable to bad reviews. Yet I would appreciate the community tending towards providing performance in these low FPR regimes since 1) current schemes are wholly capable of reaching them and 2) they are the regime of interest to the practitioner.
> >
> > The authors indicate that it is costly to compute them, this is not necessarily true:
> >
> > > On p-value in baselines
> >
> >  The reason I pointed the authors to this paper is because using exact p-values allows one to not rely on computing  watermarking performance on say 10^6 texts to report results at low FPR. Indeed, given a distribution of p-values on say a 100 texts,  I can already compute quantiles on this distribution with good accuracy. If the p-values are concentrated around 10^{-10}, I would be able to report TPR @ FPR=10^{-10} with only these 100 samples. This is why working with exact p-values -- which are available for at least KGW, Aaronson, WaterMax and SynthID -- is highly beneficial: it allows exact control of the FPR and computation of the TPR @ low FPR with a small computationnal cost.

---

> > > ### Author Response · Authors · 2025-08-06
> > >
> > > We sincerely thank the reviewer for their thoughtful feedback and for acknowledging our previous response.
> > >
> > > Regarding the first follow-up comment, we agree that the ML watermarking community should gradually shift toward evaluation metrics that better reflect the needs of real-world applications—particularly in the low-FPR regime. In future versions, we will aim to include more comprehensive experiments that report TPR at these practically important FPR thresholds.
> > >
> > > On the question of $p$-values, we appreciate the reviewer’s clarification. Our understanding is that in the cited works, the test statistics are **i.i.d.**, making it feasible to compute exact $p$-values and thus estimate low FPR quantiles with limited samples. In contrast, our detector uses **non-i.i.d.** token-level statistics due to its adaptivity to the LLM’s generative distribution, making it analytically intractable to compute exact $p$-values.
> > >
> > > To address this challenge, we provide Lemma 3 (detailed in Appendix H), which gives an upper bound on the **worst-case** sequence-level FPR: $\binom{T}{\lceil T\lambda \rceil}\eta^{\lceil T\lambda \rceil}$.
> > > This bound holds across all unwatermarked texts without distributional assumptions. As shown in Table 7 (Appendix K), this theoretical bound tightly matches the empirical FPR observed in practice.
> > >
> > > That said, if we choose $\eta$, $T$, and $\lambda$ such that the bound is  $\leq 10^{-10}$, we can, in principle, guarantee FPR $\leq 10^{-10}$ and report the corresponding TPR on as few as 100 samples. However, we chose not to report TPR in such setups, as the resulting TPR estimates may be unstable with so few positive samples and this evaluation setup is not yet widely adopted. Instead, we focused on comparing theoretical and empirical FPRs to demonstrate the reliability of our control mechanism.
> > >
> > > In the future, we will explore approximating the exact p-values of our detector and evaluate whether reporting TPR on a small number of samples provides a reliable estimate of low-FPR performance.
> > >
> > > We hope our responses have addressed your concerns, and we would be grateful if you would consider updating your score to further support our submission. Thank you again for your constructive review.

---

> > > > ### Author Response · Authors · 2025-08-09
> > > >
> > > > Dear Reviewer uUH1,
> > > >
> > > > As the deadline is approaching, we would greatly appreciate your feedback on our responses to your follow-up question. If our replies have satisfactorily addressed your concerns, we would be grateful if you could kindly confirm this and consider enhancing your score to further support our submission.
> > > >
> > > > Thank you very much for your thoughtful review and valuable time.
> > > >
> > > > Best regards,
> > > >
> > > > Paper17606 Authors

---

### Official Review · Reviewer_RWio · 2025-07-03

**Clarity:** 3
**Significance:** 2
**Originality:** 3
**Rating:** 4
**Confidence:** 4

**Summary:**

This paper presents a theoretical framework for watermarking large language models that jointly optimizes both the watermarking process and the detection method. Unlike previous approaches based on heuristics, the authors provide closed-form solutions that show the optimal watermark should adapt to the model’s natural output distribution. Building on this insight, they propose a practical algorithm called Distribution Adaptive Watermarking Algorithm, which embeds watermarks without distorting text and enables detection using a surrogate language model and the Gumbel max sampling method. Experiments on Llama and Mistral models demonstrate strong performance under low false positive rates and robustness against token-level attacks, while preserving text quality.

**Questions:**

Refer to the weakness part.

**Ethical Concerns:**

["NO or VERY MINOR ethics concerns only"]

**Final Justification:**

**The response from the authors satisfactorily resolved the issues I raised. I have accordingly increased my score upward.**

**Limitations:**

yes

**Quality:**

2

**Strengths And Weaknesses:**

Strengths:
- The paper provides a theoretically grounded framework for LLM watermarking, addressing both watermark generation and detection in a unified optimization framework.
- The distribution-adaptive approach achieves good performance at ultra-low False Positive Rates (FPR), with minimal distortion to text quality.
- Thorough experiments demonstrate the effectiveness and efficiency of DAWA on multiple datasets and language models.

Weaknesses:
- **Outdated Baseline Comparisons:** The paper lacks comparisons with recent and more competitive methods that explore the performance-detectability tradeoff, e.g. [1,2,3]. These more recent works could provide stronger baselines and highlight the significance of the proposed method.

- **Limited Discussion of Related Work**: While the paper cites many earlier watermarking techniques, it fails to address key advancements in performance-detectability tradeoff optimization from recent years. This omission weakens the contextual positioning of the proposed method.

Reference:

[1] CoheMark: A Novel Sentence-Level Watermark for Enhanced Text Quality (2025).

[2] WatME: Towards Lossless Watermarking Through Lexical Redundancy (ACL 2024).

[3] From Trade-off to Synergy: A Versatile Symbiotic Watermarking Framework for Large Language Models (2025).

---

> ### Author Rebuttal · Authors · 2025-07-31
>
> >**Q1.** Outdated Baseline Comparisons: The paper lacks comparisons with recent and more competitive methods that explore the performance-detectability tradeoff, e.g., [1,2,3]. These more recent works could provide stronger baselines and highlight the significance of the proposed method.
>
> **A1.** Among the cited works, [1] and [3] appeared **concurrently with our submission**, and thus, making a direct comparison infeasible at the time of our experiments. We will discuss them in the related work section of the final version.
>
> Regarding [2], we have conducted additional experiments to include a comparison with it. The results are summarized in the table below. Our method demonstrates favorable performance in terms of both robustness and detectability under comparable conditions.
>
> **Table 1: Clean Text**
> | Method | ROC-AUC | TP@1%FP | TP@10%FP |
> |:------:|:-------:|:------:|:-------:|
> |  WatME |  0.983  |  0.970 |  0.995  |
> |  Ours  |  0.999  |  0.998 |  1.000  |
>
> **Table 2: Token Replacement Attack**
> | Method | ROC-AUC | TP@1%FP | TP@10%FP |
> |:------:|:-------:|:------:|:-------:|
> |  WatME |  0.923  |  0.623 |  0.750  |
> |  Ours  |  0.989  |  0.860 |  0.976  |
>
>
> ---
> >**Q2.** Limited Discussion of Related Work: While the paper cites many earlier watermarking techniques, it fails to address key advancements in performance-detectability tradeoff optimization from recent years. This omission weakens the contextual positioning of the proposed method.
>
> **A2.** To the best of our knowledge, among the plethora of LLM watermarking works, we have cited the most relevant and up-to-date ones both from the empirical and theoretical perspectives. For a more comprehensive literature review, we refer the readers to Appendix A and the recent surveys [4,5] ([2,3] in our manuscript), due to the space limit.
>
> If there are specific recent works that we have overlooked, we would appreciate it if the reviewer could point them out. We will make sure to include them and expand the related work section in the final version.
>
>
> **Reference:**
>
> [4] Aiwei Liu, Leyi Pan, Yijian Lu, Jingjing Li, Xuming Hu, Xi Zhang, Lijie Wen, Irwin King, Hui Xiong, and Philip Yu. A survey of text watermarking in the era of large language models. ACM Computing Surveys, 57(2):1–36, 2024.
>
> [5] Junchao Wu, Shu Yang, Runzhe Zhan, Yulin Yuan, Lidia Sam Chao, and Derek Fai Wong. A survey on LLM-generated text detection: Necessity, methods, and future directions. Computational Linguistics, pages 1–66, 2025.

---

### Official Review · Reviewer_7FZT · 2025-07-05

**Clarity:** 3
**Significance:** 4
**Originality:** 3
**Rating:** 5
**Confidence:** 3

**Summary:**

The paper addresses the critical problem of deriving the optimal watermarking for LLM. By framing watermarking as a hypothesis testing under distortion constraint, it derives a unified closed-form paradigm of the optimal watermarking scheme. The paper further draws interesting theoretical insights from the detection error-distortion tradeoff, buliding on which it develops a practical token-level watermarking protocol. Empirical evaluations confirms the superiority of the proposed scheme.

**Questions:**

How does the proposed optimal watermarking scheme reduces to the likelihood-ratio test for a fixed P vs. Q?
Could you provide concrete examples of how existing watermarking schemes instantiate the bijection g_{t,k}, showing each as a special case of your general mapping function?

**Ethical Concerns:**

["NO or VERY MINOR ethics concerns only"]

**Final Justification:**

My question "the paper does not explicitly discuss how the proposed optimal detector reduces to the canonical ratio test" has been addressed to resolve my misunderstanding. Therefore I'd like to raise the score.

**Limitations:**

yes

**Paper Formatting Concerns:**

No major formatting issues are spotted.

**Quality:**

3

**Strengths And Weaknesses:**

Strengths: The paper casts warermarking as a fresh constrained hypothesis-testing problem, which effectively unifies the watermark embedding and detection into one optimality framework. It manages to give a closed-form expression for the Type-I Type-II tradeoff, with optimality provable from both theoretical and empirical perspectives. Moreover, it provides a plug-and-play paradigm, where any concrete watermark design can be adapted accordingly to yield its own provable optimal detector, making the framework highly general.

Weakness: The paper leverages the classical hypothesis testing framework, which inherently has an optimal rejection region via the likelihood-ratio test as guaranteed by the Neymann-Pearson lemma. However, the paper does not explicitly discuss how the proposed optimal detector reduces to the canonical ratio test. In addition, the core derivations are rigorous but somewhat abstract and dense with little high-level intuitions, making the manuscript less readable.

---

> ### Author Rebuttal · Authors · 2025-07-31
>
> >**Q1.** The paper does not explicitly discuss how the proposed optimal detector reduces to the canonical ratio test. How does the proposed optimal watermarking scheme reduce to the likelihood-ratio test for a fixed P vs. Q?
>
> **A1.** We appreciate the reviewer’s question. The canonical likelihood-ratio test (LRT) is optimal when the null and alternative hypotheses are fully specified by fixed distributions $P$ and $Q$, and when the goal is to minimize the Type-II error (1-TPR) subject to a constraint on the Type-I error (FPR). However, our setting is fundamentally different.
>
> In our formulation:
>
> 1. **Neither** $H_0$ nor $H_1$ corresponds to a fixed distribution: Under $H_0$, the distribution $P_{\zeta_1^T}$ over auxiliary sequences is **induced by the watermarking scheme** and is **unspecified**. Under $H_1$, the joint distribution $P_{X_1^T, \zeta_1^T}$ **depends on the design** of the watermarking scheme and is **to be optimized**. Therefore, our detection task is a composite hypothesis test, and the LRT is not optimal in this setting.
> 2. Our goal is to minimize the Type-II error (1-TPR) subject to a constraint on the **worst-case FPR** and a constraint on the **watermarked text distortion**.
> 3. Moreover, as emphasized in Line 33, **the detector is required to be model-agnostic**, i.e., it cannot depend on $P_{\zeta_1^T}$, $P_{X_1^T, \zeta_1^T}$, or $Q_{X_1^T}$, as these distributions may not be available to the detector at test time. Therefore, the likelihood ratio test, which requires knowledge of the underlying distributions, is not applicable.
>
> These key constraints distinguish our problem from classical hypothesis testing. Given this setup, Theorem 2 characterizes the **jointly optimal watermarking scheme and detector**, where the optimal detector is determined by some chosen function $g$. Notably, this detector **does not reduce to an LRT**, precisely because the assumptions and objectives of the classical setting do not apply in our framework.
>
> ---
>
> >**Q2.** Could you provide concrete examples of how existing watermarking schemes instantiate the bijection $g_{t,k}$, showing each as a special case of your general mapping function?
>
> **A2.** We clarify that our framework characterizes the jointly optimal pairs of detector and watermarking scheme. As a result, the existing watermarking scheme, many of which rely on heuristics or focus solely on optimizing the detector, are not special cases of the jointly optimal design and do not instantiate our derived optimal detector.
>
> In our practical detector design (Eq. (5)), the bijection function $g_{tk}$ is a token-level adaptation of the bijection function $g$ from Theorem 2. To offer a concrete toy example: suppose the token vocabulary is $\mathcal{V}=\lbrace a,b,c \rbrace$. We can define an extended set $\mathcal{S}=\lbrace a,b,c,\\# \rbrace$ $\supset \mathcal{V}$ and a bijective $g_{tk}$ that maps auxiliary symbols $\mathcal{Z}=\lbrace 1,2,3,4 \rbrace$ to elements in $\mathcal{S}$.
>
> ---
>
> >**Q3.** The core derivations are rigorous but somewhat abstract and dense, with little high-level intuition, making the manuscript less readable.
>
> **A3.** We acknowledge that the core derivations are mathematically involved, and we have aimed to provide high-level intuition through the visual illustrations in Figures 2, 3, and 5, which highlight the theoretical insights and trade-offs. We will revise the manuscript to further improve clarity, including adding more intuition and commentary around key results, to make the presentation more accessible in the final version.
>
>
> ---
> We sincerely appreciate your time and valuable feedback. We hope our responses have adequately addressed your concerns, and if so, we would be grateful if you could consider revising your score to further support our submission. Thank you again for your thoughtful review.

---

> > ### Comment · Reviewer_7FZT · 2025-08-08
> >
> > I thank the authors for the clear rebuttal—it helped resolve my earlier misunderstandings, and I’m likely to raise my score.
> >
> > I’d encourage the authors to fold the toy example in A2 on the bijection function into the main manuscript, since several reviewers stumbled there on first read. Likewise, it would be helpful to add a short discussion around A1 to highlight the fundamental difference between your hypothesis-testing setup and the classic one, as clarified in the rebuttal. Making these points explicit in the paper will improve clarity for future readers. Looking forward to seeing these updates in the revised draft.

---

> > > ### Author Response · Authors · 2025-08-08
> > >
> > > We sincerely thank the reviewer's insightful suggestions and the upgraded score. We will integrate the advice into our finalized manuscript. Thank you for your further support!

---

### Decision · Program_Chairs · 2025-09-17

**Decision:**

Accept (poster)

**Comment:**

This paper studies watermarking of LLM outputs from a hypothesis testing perspective. The authors present a theoretical formulation that aims to optimize both the watermarking and detection through this lens. The paper obtains useful insights from this formulation that, in particular, proves (the perhaps intuitive fact) that an optimal algorithms should adapt to the LLM’s algorithm (which is actually very intuitive).

The paper then moves to use ideas and insights from their theoretical framework to design an algorithm for practice that they call the distribution-adaptive watermarking algorithm (DAWA). This scheme is optimal at the token level but it also inherits some benefits at the sentence level.

Overall, the reviewers appreciated the theoretical formulation (more) but also found the DAWA useful as well. On the down side, the practical solution makes several compromises and its robustness is also quite limited. The reviewers thought that the positive side outweighs the downside and hence the paper can be a nice addition to the NeurIPS program.